# CITED2 is a druggable epigenetic switch coupling neuronal maturation to regenerative decline

Franziska Müller [ID][1], Eilidh McLachlan[1], Ana Catarina Costa[2], Jia Qu[3], Bishal Shrestha [ID][4], Zheng Wang[4], Francesco De Virgiliis[5], Thomas Haynes Hutson[6], Luming Zhou[1], Guiping Kong[1], Jessica S Chadwick[1], Paolo La Montanara [ID][1], Zhulin Yuan[1], Nejc Haberman[1], Monica M Sousa[2], Ilaria Palmisano [ID][3] & Simone Di Giovanni [ID][1][✉]

## Abstract

**Neuronal maturation involves a tightly regulated cessation of growth and acquisition of polarity, ultimately leading to synapse formation. While essential for circuit stability, maturation marks the loss of regenerative capacity after central nervous system (CNS) injury. The molecular programs coupling maturation to regenerative decline remain incompletely understood. Here, we show that the transcriptional and epigenetic signatures enabling axon growth in dorsal root ganglion (DRG) neurons are lost as they transition from immature, non-polarized cells to mature, pseudo-unipolar neurons. We identify the transcriptional co-regulator CITED2 as a key epigenetic switch, active in immature and regenerating DRG neurons but silent after non-regenerative spinal cord injury (SCI). Cited2 overexpression reactivates growth programs, enhancing regeneration in vivo after SCI. Mechanistically, CITED2 reinstates developmental epigenetic and transcriptional profiles, decoupling maturation from regenerative failure. Pharmacogenomic screening identified CITED2 as a target of the clinically approved HDAC inhibitor Panobinostat, which promoted axonal growth, sprouting, and functional recovery post-injury. These findings position CITED2 as a key regulator of sensory neuron plasticity and a novel therapeutic target for CNS repair.**

**Keywords** Neuronal Maturation; Cited2; Axonal Regeneration; Spinal Cord Injury; Transcription
**Subject Category** Neuroscience

## Introduction

The decline in regenerative potential during neuronal maturation is driven by coordinated changes in gene expression and chromatin architecture. Previous studies have shown that this transition involves widespread epigenetic remodeling, including alterations in chromatin accessibility, histone acetylation, and 3D genome organization (Gallegos et al, 2018; Ciceri et al, 2024; Luo et al, 2021). In particular, growth-associated genes are progressively repressed via chromatin compaction and rewiring of chromatin loops to favor gene programs required for polarization, synaptogenesis, and functional maturation (Palmisano et al, 2019; Palmisano et al, 2024; Rahman et al, 2023; Venkatesh et al, 2016).

Despite these insights, key questions remain unresolved: (1) Does the epigenetic silencing of axon growth potential occur progressively across neuronal maturation, or is it executed as a discrete, switch-like transition at defined developmental stages? (2) What regulators orchestrate this transition? (3) Can these regulators be manipulated to restore growth in mature neurons?

The dorsal root ganglion (DRG) sensory system offers an ideal model to address these questions. DRG neurons undergo a well-defined maturation process, transitioning from a spindle-shaped bipolar morphology to a polarized pseudo-unipolar structure between embryonic days E12.5 and E17.5. This transition is marked by dramatic cytoskeletal and transcriptional reprogramming that culminates in the cessation of axon growth and the onset of synapse formation (Takahashi and Ninomiya, 1987; Barber and Vaughn, 1986; Prasad and Weiner, 2011; Nascimento et al, 2018; Tedeschi et al, 2016) Importantly, this developmental process can be recapitulated in vitro (Nascimento et al, 2022).

DRG neurons display a unique regenerative dichotomy: peripheral sciatic nerve injury triggers robust regenerative programs, while central spinal cord injury (SCI) does not. Our previous work demonstrated that this divergence is underpinned by distinct epigenetic landscapes. Using machine learning to integrate RNA-seq and chromatin profiling after peripheral versus central lesions, we identified gene expression and histone acetylation patterns predictive of regenerative capacity (Palmisano et al, 2019). Regenerative injury selectively enhances chromatin accessibility and recruits developmental transcriptional regulators. More recently, we found that it also remodels 3D genome organization

[1]Division of Neuroscience, Department of Brain Sciences, Imperial College London, London W12 0NN, UK. [2]Nerve Regeneration Group, Instituto de Biologia Molecular e Celular (IBMC), Instituto de Investigação e Inovação em Saúde (i3S), University of Porto, Porto 4200-135, Portugal. [3]Department of Neuroscience, Ohio State University College of Medicine, Columbus, OH 43210, USA. [4]Department of Computer Science, University of Miami, Coral Gables, FL 33124-4245, USA. [5]Department of Pathology and Immunology, University of Geneva, Geneva 1211, Switzerland. [6]The Wyss Center for Bio and Neuroengineering in Geneva, Chemin des Mines 9, 1202 Geneva, Switzerland.
[✉]E-mail: s.di-giovanni@imperial.ac.uk

in DRG neurons, extending chromatin loops that link distal enhancers to pro-growth gene promoters (Palmisano et al, 2024).

Here, we compared regenerative gene expression, epigenetic, and chromatin architecture profiles after injury with gene expression signatures associated with DRG maturation. This revealed CITED2 (Cbp/P300-interacting transactivator with Glu/Asp-rich carboxy-terminal domain 2) as a key transcriptional cofactor selectively highly expressed in non-polarized immature DRG neurons and reactivated following regenerative sciatic injury, which, however, remains unchanged after SCI. Cited2 overexpression in cultured DRG neurons reinitiated a growth-competent state, while Cited2 knockdown suppressed regenerative growth. In vivo, AAV-mediated overexpression of CITED2 in adult DRG neurons enhanced axon regeneration after SCI. Mechanistically, RNA-seq and ATAC-seq profiling showed that CITED2 restored a transcriptional and epigenetic state akin to early developmental stages, marked by chromatin decompaction, histone acetylation, and accessibility of regeneration-associated gene loci.

Together, our findings identify a critical developmental window, between E12.5 and E17.5, during which sensory neurons transition from a growth-competent to a polarized, post-mitotic state; importantly, this process is reversible. CITED2 acts as a molecular switch capable of reprogramming mature neurons into a regenerative mode by reinstating chromatin plasticity and developmental gene expression programs.

These results uncover a reversible epigenetic brake on axon growth and point to CITED2 as a therapeutic target to restore regenerative capacity in the injured adult nervous system. Furthermore, targeting Cited2 expression by using the clinically approved small-molecule HDAC-inhibitor Panobinostat after SCI promoted neuronal growth, sprouting, and functional recovery.

## Results

### The transition from an immature non-polarized to a mature polarized stage marks the loss of gene expression signatures associated with the regenerative growth competence of DRG neurons

We first investigated whether the regenerative growth competence of adult mouse DRG neurons is associated with gene expression signatures specific of immature non-polarized DRG neurons, which are lost during neuronal maturation (Fig. 1A). To this end, we examined the association between differential gene expression profiles after a regenerative competent sciatic nerve axotomy (SNA) or regenerative incompetent spinal cord dorsal column axotomy (DCA) with sequential stages of neuronal maturation (E12.5: neuronal growth, E14.5: intermediate, E17.5: synapse formation) during embryonic development (Dataset EV1) by using previously published mouse DRG RNA-seq datasets (Data ref: Southard-Smith, et al, 2014; Tedeschi et al, 2016; Data ref: Tedeschi et al, 2016; Palmisano et al, 2019; Data ref: Hervera et al, 2018) Strikingly, this analysis revealed a significant association between differential gene expression profiles represented during the transition between E12.5 and E17.5 and the regenerative competent SNA versus sham (Fig. 1B). A less significant association was also observed between the regenerative SNA versus sham and the transition between E14.5 and E17.5 (Fig. 1B). No significant

relationship was identified between neuronal maturation and regenerative incompetence (DCA vs laminectomy, LAM) or between E12.5 versus E14.5 and SNA versus sham (Fig. 1B). Specifically, we found that 190 upregulated and 205 downregulated genes were shared between the developmental stage E12.5 and a regenerative competent SNA versus sham (Fig. 1C,D; Dataset EV2). Highly ranked biological functional classes of upregulated genes included regulation of transcription, cell cycle, apoptosis, and phosphorylation; while among the downregulated genes, terms included ion and membrane transport, and synaptic regulation (Fig. 1E; Dataset EV2).

This analysis suggests that the loss of regenerative growth potential might be marked by the transition of DRG neurons to a polarized morphology.

### Cited2 expression is induced by a regenerative competent peripheral nerve injury and supports axon growth of DRG neurons, including after SCI

Gene expression dynamics during development are accompanied by coordinated changes in epigenetic signatures that ensure proper activation of developmental programs (Ciceri et al, 2024; Joseph et al, 2021; Luo et al, 2021; Ding et al, 2016). We recently demonstrated that epigenetic landscapes and chromatin accessibility, including those at developmentally regulated loci, can predict the regenerative growth potential of sensory DRG neurons following injury (Palmisano et al, 2019). To identify key epigenetic regulators driving the transition from immature, non-polarized neurons to mature, polarized states potentially underlying the decline in regenerative capacity, we examined genes shared between E12.5 versus E17.5 developmental stages and SNA versus sham conditions. We analysed their transcriptional and epigenetic profiles after a regenerative peripheral injury using previously published datasets from whole DRG and DRG neuronal nuclei (Data ref: Southard-Smith et al, 2014; Tedeschi et al, 2016; Data ref: Tedeschi et al, 2016; Palmisano et al, 2019; Palmisano et al, 2024; Data ref: Palmisano et al, 2024; Data ref: Hervera et al, 2018). Specifically, we integrated differential gene expression (RNA), chromatin accessibility (ATAC), histone acetylation (H3k9ac and/or H3k27ac), and their localization within cohesin-dependent topologically associated domains, which we recently identified as essential for successful axon regeneration (Palmisano et al, 2024) (Fig. 2A).

This integrative analysis identified CITED2 (Cbp/P300-interacting transactivator with Glu/Asp-rich carboxy-terminal domain 2) (Bhattacharya et al, 1999), a transcriptional co-activator and key developmental regulator, as a prime epigenetic candidate. CITED2 exhibited a coordinated upregulation of gene expression, enhanced chromatin accessibility, increased H3 acetylation, and dependence on cohesin-mediated 3D chromatin architecture (Fig. 2A). Notably, Cited2 mRNA levels peaked both at the immature, non-polarized E12.5 stage and after regenerative conditioning following SNA (Fig. 2B).

Next, we aimed to attest whether Cited2 expression was indeed selectively induced by SNA. Immunofluorescence (Fig. 2C–F), RT-qPCR (Fig. 2G), and immunoblotting (Fig. 2H–J) confirmed that Cited2 was significantly upregulated at the mRNA and protein level in DRG neurons after a regenerative competent SNA but not after a regenerative incompetent DCA.

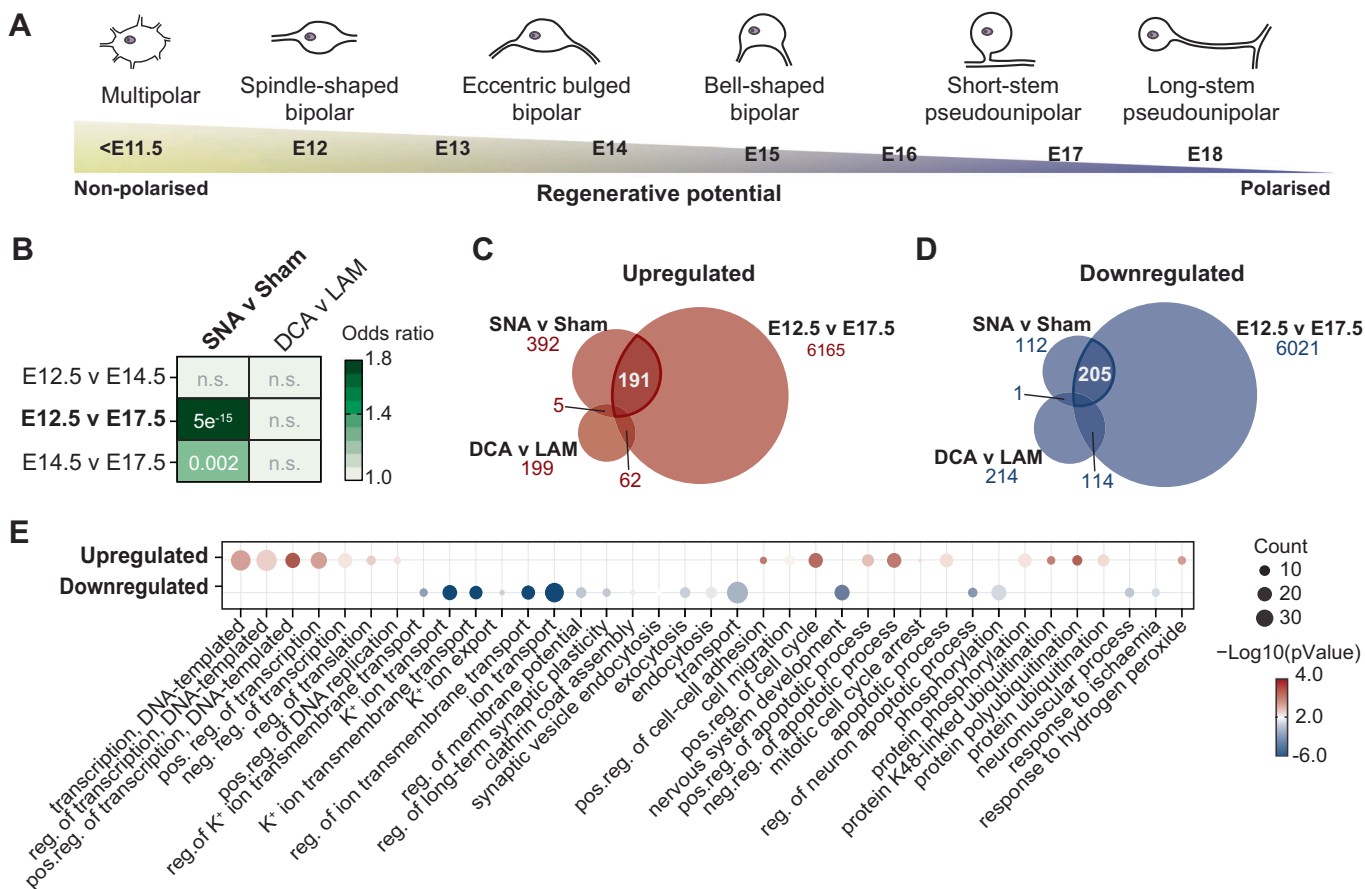

**Figure 1. The developmental transition from an immature non-polarized to a mature polarized state in DRG neurons marks the loss of gene expression signatures associated with growth competence.**

(A) Schematic describing the transition from an immature non-polarized to a mature pseudo-unipolar phenotype during DRG neuronal maturation and the hypothetical association with growth ability underpinning the regenerative potential. (B) Odds ratio and Fisher's exact test of differentially expressed genes (FDR <0.05) from RNA-seq datasets comparing the regenerative injury (SNA v sham) or non-regenerative injury (DCA v LAM) to developmental timepoints, reflecting a progressively declining neuronal growth ability (E12.5 > E14.5 > E17.5). Color represents correlation (odds ratio) while numbers represent significance (Fisher's exact test). (C, D) Upregulated and downregulated genes shared between SNA v sham, E12.5 v E17.5, and DCA v LAM. (E) Gene ontology (GO) and KEGG analysis of shared upregulated and downregulated genes between SNA v sham and E12.5 v E17.5. P values are calculated using a modified Fisher's exact test as described in DAVID (Huang et al, 2009). Source data are available online for this figure.

To determine if Cited2 was required and sufficient for regenerative growth of DRG neurons, both loss and gain of function experiments were performed *in culture*. Briefly, cultured adult DRG neurons were electroporated with GFP and CRISPR-Cas9 Cited2 double nickase to knock down Cited2, or GFP and CRISPR-Cas9 double nickase control plasmids. Quantification of mean neurite outgrowth revealed that Cited2 knockdown, which reduced Cited2 expression (Fig. EV1A), significantly impaired neurite outgrowth with respect to control-transfected cells (Fig. 3A,B). Adult DRG neurons were electroporated with either control GFP or Cited2-GFP expressing plasmids to overexpress Cited2 and cultured on growth-permissive (Poly-D-lysine and laminin) or inhibitory (myelin) substrates. GFP control or Cited2 plasmid overexpression was confirmed using immunofluorescence (Fig. 3C,D) and immunoblotting (Fig. 3E,F). A significant increase in average neurite length on both permissive (Fig. 3G,H) and inhibitory (Fig. 3I,J) substrates was found in Cited2 overexpressing neurons.

Finally, we overexpressed either control AAV-GFP or AAV-Cited2-GFP in DRG neurons in vivo in mice. Cited2 over-expression was first confirmed in DRG cultured neurons by immunoblotting (Fig. EV1B,C) and immunofluorescence (Fig. EV1D,E). Next, the AAV vectors were injected into the sciatic nerve bilaterally, 4 weeks prior to a T9 spinal cord DCA (Fig. 4A). At 6 weeks post-injury, we found a significant increase in sensory axon growth into and past the lesion site in mice overexpressing Cited2 (Figs. 4B,C and EV2). Both a significant increase in axon growth rostral to the lesion site and a significant reduction in axon retraction were observed following Cited2 overexpression in comparison to the GFP control group (Fig. 4D). As expected, AAV-mediated Cited2 overexpression resulted in a significant increase in Cited2 expression in DRG neurons 6 weeks post-SCI (Fig. 4E, F). Cited2 overexpression also increased EP300 and active CBP (acetylated CBP; acCBP) expression (Appendix Fig. S1A–D) and augmented the acetylation of H3K9 and H3K27 (Appendix Fig. S1E–H), which are associated with an increased regenerative state. We also determined that AAV-Cited2-GFP overexpression did not affect the lesion size or GFAP expression within the injury area compared to the AAV-GFP control (Appendix Fig. S2A–C).

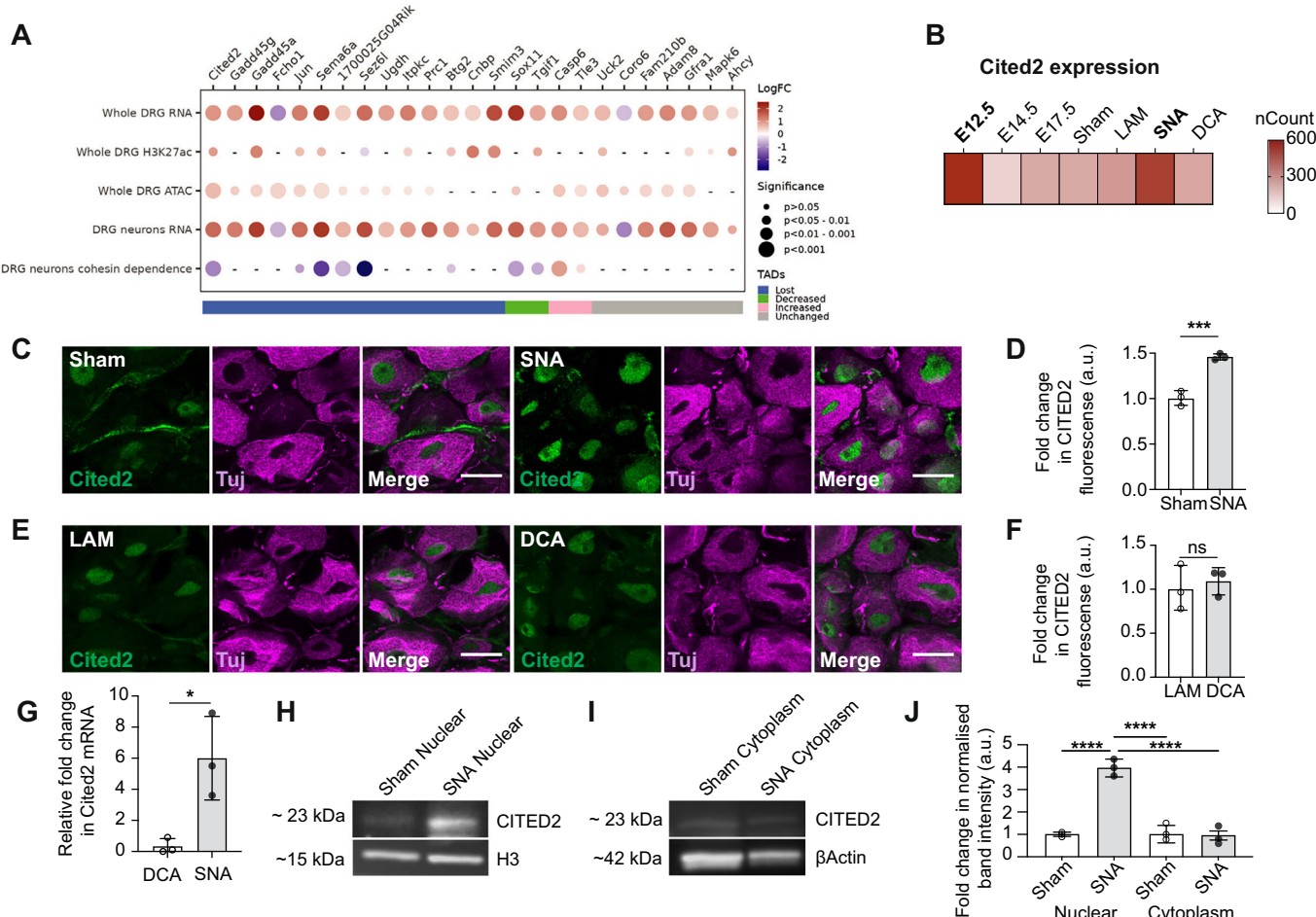

**Figure 2. Cited2 is identified as a key factor involved in the transition from E12.5 to E17.5 during neuronal maturation and is induced after regenerative injury.**

(A) Dot plot of upregulated and downregulated genes in DRG and DRG neurons after SNA v sham and during developmental transition from E12.5 v E17.5, showing changes in chromatin accessibility (ATAC), histone H3k27 acetylation (H3K27ac) at the promoter and/or gene body and that rely on cohesin-dependent topologically associated domains (TADs). Dot size indicates significance, color indicates Log Fold Change. (B) Normalized transcript per million (TPM) counts for Cited2 across different stages of DRG development and injury conditions. (C, D) CITED2 immunostaining (green) in DRG neurons (beta 3 tubulin; TUJ) 24 h after SNA or sham injury with quantification of fold change (t-test, $p = 0.0003$, $n = 3$ independent biological replicates, DRG neurons analysed throughout the thickness of the sciatic DRG). (E, F) CITED2 immunostaining (green) in DRG neurons (TUJ) 24 h after DCA or LAM injury with quantification of fold change (t-test, $p = 0.7044$, $n = 3$ independent biological replicates, DRG neurons analysed throughout the thickness of the sciatic DRG). (G) Fold change in Cited2 mRNA expression normalized to beta-actin, assessed by qPCR analysis from sciatic DRGs 24 h after SNA v sham or DCA v LAM injuries (t-test, $p = 0.0229$, $n = 3$ independent biological replicates). (H–J) Western blot analysis from nuclear and cytoplasmic lysates, extracted from sciatic DRG dissected 24 h after sham or SNA. Nuclear and cytoplasmic band intensity normalized against H3 and beta-actin, respectively ($p = 3.007 \times 10^{-6}$, by one-way ANOVA with Tukey's post hoc test, $n = 3$ independent biological replicates). All scale bars: 50 μm. *$p < 0.05$, **$p < 0.01$, ***$p < 0.001$, and ****$p < 0.0001$. ns = not significant. All error bars shown as standard deviation (SD). Source data are available online for this figure.

Together, these data demonstrate that Cited2 is highly expressed in immature DRG neurons as well as after regenerative SNA. Furthermore, loss- and gain-of-function in cultured DRG neurons and overexpression in vivo show a role for Cited2 in axonal growth of DRG neurons, including after a SCI.

## Cited2 overexpression in DRG neurons promotes gene expression and epigenetic signatures associated with a neuronal immature state after SCI

Changes in gene expression and chromatin accessibility are prominent following a regenerative peripheral injury and are essential for the coordinated control of neuronal development and

maturation (Oh et al, 2018; Palmisano et al, 2019; Palmisano et al, 2024; Hilton et al, 2024). Given CITED2 is a transcriptional co-regulator, we next examined the gene expression programs associated with its in vivo overexpression in adult DRG neurons after SCI. To this end, AAV-GFP or AAV-Cited2-GFP was bilaterally injected into the sciatic nerve 4 weeks before a regeneration-incompetent T9 dorsal column axotomy (DCA). Twenty-four hours after injury, L4–L6 DRG were dissected, and dissociated cells were enriched for neurons using a 15% BSA cushion (Fig. EV3A–E)(Chadwick et al, 2025), followed by RNA-seq analysis (Fig. 5A; Dataset EV3).

RNA-seq analysis indicated 908 differentially expressed genes (DEGs), of which 420 and 488 were significantly ($p < 0.05$)

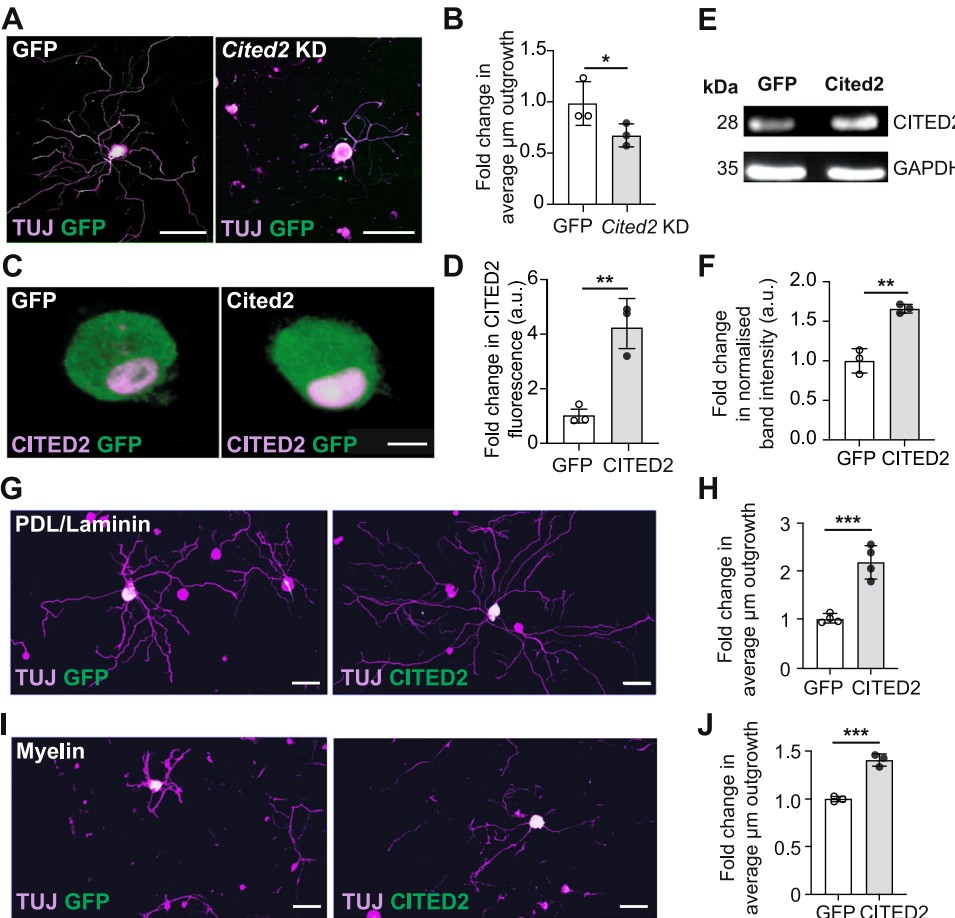

**Figure 3. Cited2 regulates DRG neuronal outgrowth.**

(A, B) Neurite outgrowth following knockdown of Cited2 in DRG neurons (TUJ, magenta. GFP marks transfected cells) with quantification of fold change in average neurite outgrowth (t-test, $p = 0.0432$, $n = 3$ independent biological replicates, at least 50 cells per replicate analysed). Scale bar: 100 µm. (C, D) CITED2 overexpression (magenta) in GFP⁺ DRG neurons (green), after electroporation with GFP or Cited2-GFP plasmids (t-test, $p = 0.0042$, $n = 3$ independent biological replicates, at least 50 cells per replicate analysed). Scale bar: 10 µm. (E, F) Immunoblotting after CITED2 overexpression in F11 cell lysates. Band intensity normalized against GAPDH (t-test, $p = 0.0022$, $n = 3$ independent biological replicates). (G–J) DRG neurite outgrowth (TUJ, magenta) after electroporation with GFP control or Cited2-GFP plasmids (green) on growth permissive (PDL/laminin) or inhibitory (myelin) substrates with quantification of the fold change in average neurite outgrowth (t-test, H: $p = 0.0005$, J: $p = 0.0006$, $n = 3$ and 4 independent biological replicates, at least 50 cells per replicate analysed). Scale bar: 100 µm. *$p < 0.05$, **$p < 0.01$, ***$p < 0.001$, and ****$p < 0.0001$. ns not significant. All error bars shown as standard deviation (SD). Source data are available online for this figure.

upregulated or downregulated, respectively (Fig. 5B). Heatmap and hierarchical clustering of DEGs indicated different gene expression clustering between Cited2 overexpression or GFP control samples (Fig. 5C). Additionally, Cited2 was indeed the most significantly upregulated gene in the dataset (Fig. 5D; Dataset EV3). Immunofluorescence also confirmed CITED2, EP300, and acCBP overexpression (Appendix Fig. S3A–F) and an increase in H3K27ac and H3K9ac (Appendix Fig. S3G–J) in DRG neurons twenty-four hours after DCA.

Following Cited2 overexpression, Gene Ontology (GO) and KEGG pathway analysis revealed that upregulated genes were significantly associated with neuronal growth, including terms such as neuron development, nervous system process, neuron projection development, and cell morphology, while downregulated genes were associated with terms including ion and neurotransmitter transport, neurotransmitter secretion, and synaptic signaling (Fig. 5E; Dataset EV3).

To further examine whether Cited2 overexpression indeed activates developmentally regulated genes and signaling pathways, we examined the relationship between Cited2 overexpression and the E12.5 immature non-polarized neuronal state. Normalized Cited2 expression levels mimicked those observed at E12.5 (Fig. 5F). Strikingly, Cited2 overexpression in DRG neurons post-SCI (Cited2 versus GFP) was significantly associated with the E12.5 immature non-polarized neuronal state (E12.5 versus E17.5) (Fig. 5G,H) and showed a stronger correlation to E12.5 compared to the regenerative sciatic nerve injury (SNA versus sham and E12.5 versus E17.5) (Fig. 5G), suggesting Cited2 overexpression may promote similar signaling pathways as observed at E12.5. No significant correlation was found between Cited2 versus GFP and E14.5 versus E17.5 (Fig. 5G).

Examining genes shared between Cited2 overexpression and the E12.5 immature non-polarized neuronal state indicated 117 upregulated and 286 downregulated common genes (Fig. 5I;

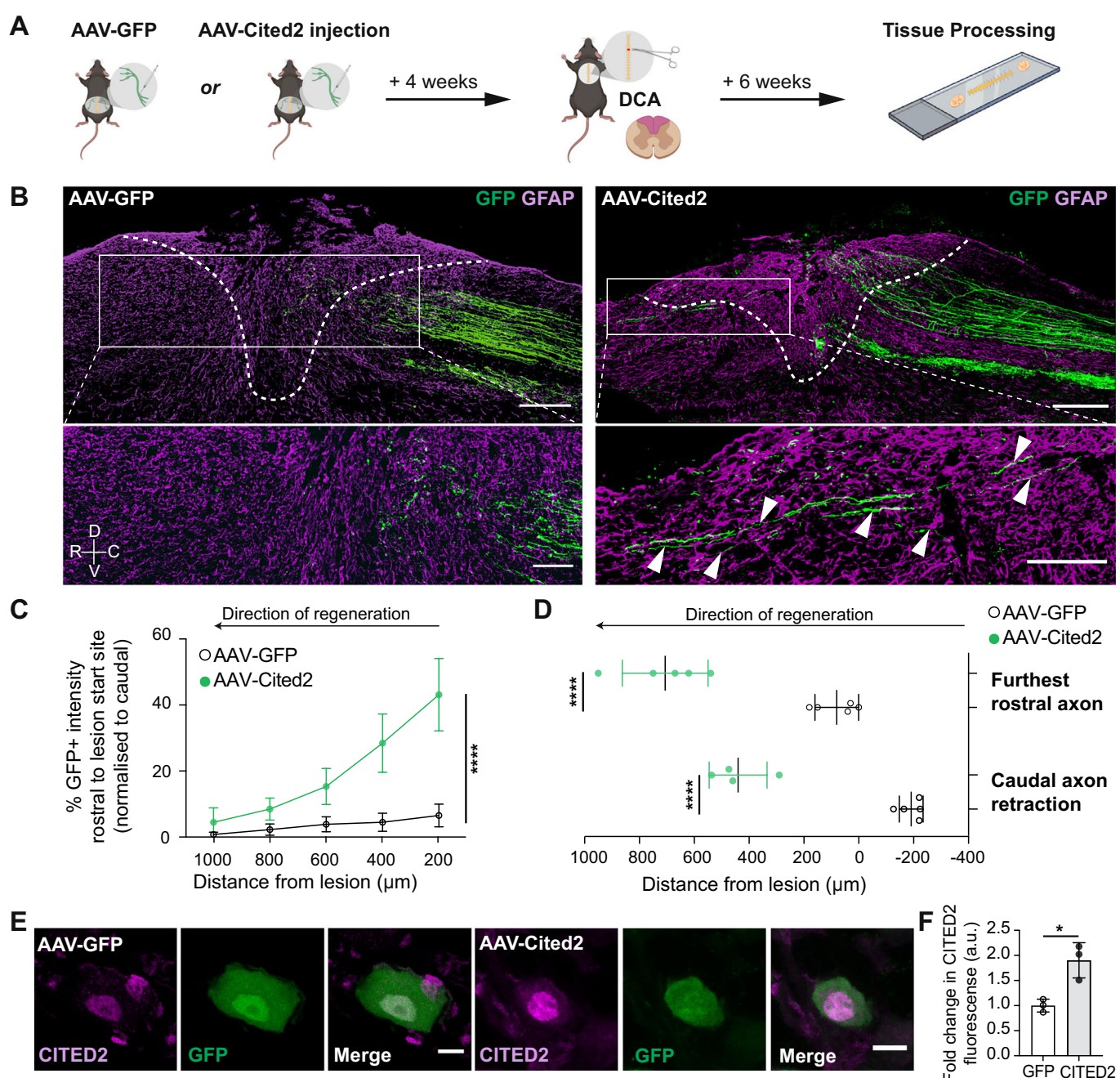

**Figure 4. AAV-mediated overexpression of CITED2 promotes axonal growth after SCI.**

(A) Experimental design (made with BioRender). (B) Representative micrographs of AAV-GFP or AAV-Cited2-GFP overexpression (green, white arrows) six weeks after SCI. GFAP (magenta) was used to determine the lesion site (white dotted line). Scale bar: large image 200 µm, zoomed image 100 µm. (C) Percentage GFP intensity rostral to the lesion border in AAV-GFP or AAV-Cited2-GFP overexpressing mice after SCI. Fluorescence intensity rostral to the lesion border was normalized to GFP intensity caudal to the lesion site. Two-way ANOVA with Sidak post hoc test; $n = 7$ independent biological replicates. Group: $f(1) = 187.94$, $p = 1.799 \times 10^{-21}$; Distance: $f(4) = 45.16$, $p = 1.282 \times 10^{-18}$; Interaction score: $f(4) = 26.44$, $p = 4.128 \times 10^{-13}$. Displayed is the Interaction $p$ value. (D) Average distance between the lesion border and the furthest rostral ($t$-test, $p = 4.4519 \times 10^{-5}$) or caudal axons ($t$-test, $p = 5.3729 \times 10^{-6}$). $n = 4$-5 independent biological replicates. (E, F) CITED2 immunostaining (magenta) in DRG neurons after AAV-GFP or AAV-Cited2-GFP overexpression (green) six weeks post-SCI ($t$-test, $p = 0.0139$, $n = 3$ independent biological replicates, DRG neurons analysed throughout the thickness of the sciatic DRG. Scale bar: 20 µm). *$p < 0.05$, **$p < 0.01$, ***$p < 0.001$, and ****$p < 0.0001$. ns not significant. All error bars shown as standard deviation (SD). Source data are available online for this figure.

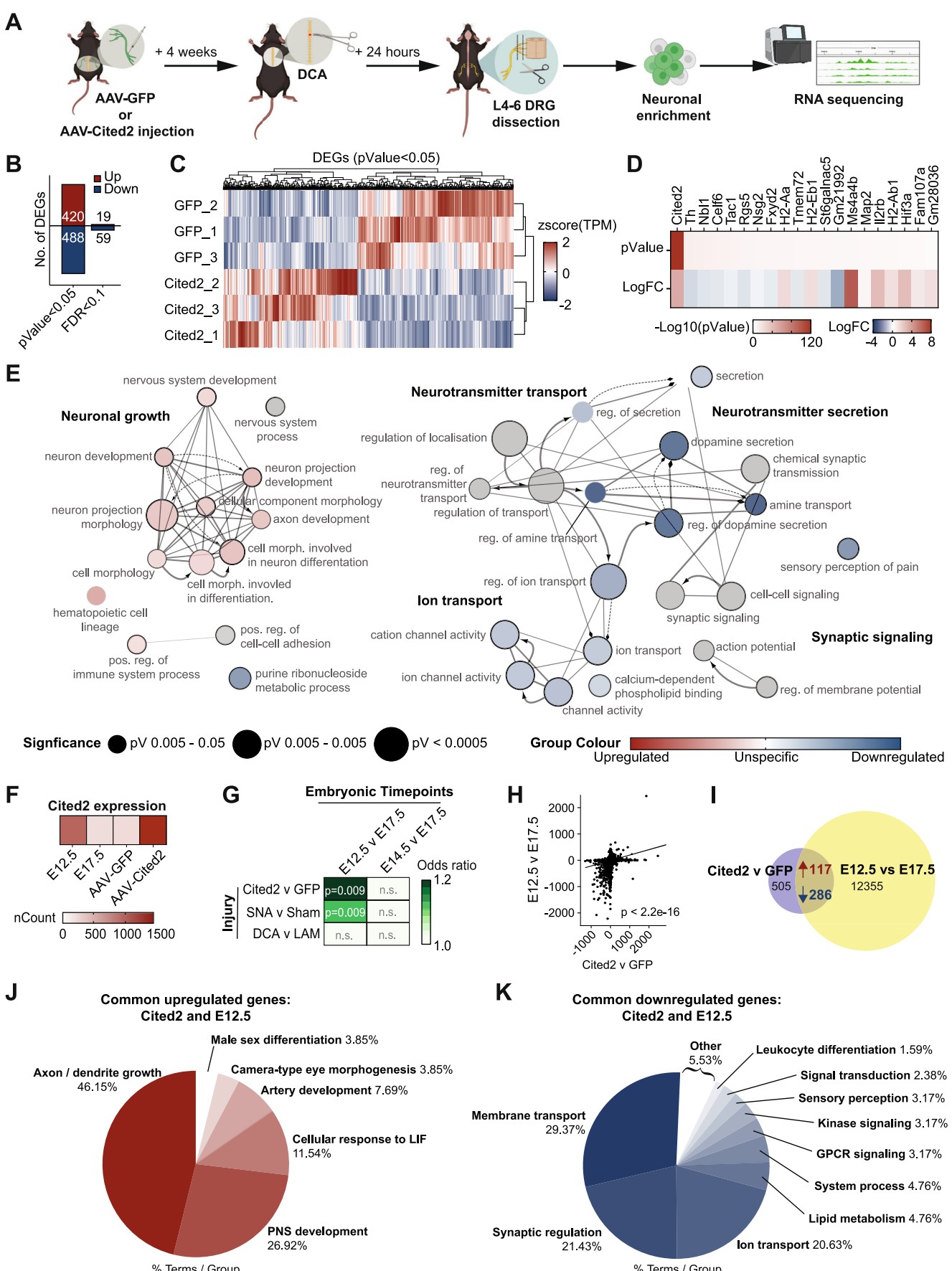

**Figure 5.   CITED2 overexpression in DRG neurons after SCI promotes gene expression signatures associated with a neuronal growth-competent immature state.**

(A) Experimental design (made with BioRender), n = 3 independent biological replicates were input for RNA-seq. (B) Number of significantly differentially expressed genes (DEGs). DEGs are identified using DESeq2, where the significance threshold is defined as p value <0.05 or FDR <0.1. For all downstream analysis, p value threshold was applied. (C) Heatmap and hierarchical clustering of DEGs (p < 0.05) from AAV-GFP control and AAV-Cited2-GFP samples. (D) Top 20 significant DEGs (p < 0.05). (E) GO and KEGG analysis of differentially expressed upregulated and downregulated genes following Cited2 v GFP overexpression. (F) Normalized TPM counts for Cited2 at E12.5 and E17.5 DRG neuron maturation stage and after AAV-GFP or AAV-Cited2-GFP overexpression. (G) Odds ratio and Fisher's exact test between Cited2 v GFP differentially expressed genes and E12.5 v E17.5, E14.5 v E17.5, and DCA v LAM. Color represents correlation (odds ratio) while numbers represent significance (Fisher's exact test). (H) Pearson correlation analysis between E12.5 v E17.5 and Cited2 v GFP ($p = 2.475 \times 10^{-7}$; $R = 0.25$). (I) Number of significantly DEGs shared between Cited2 v GFP and E12.5 v E17.5 datasets. (J, K) GO and KEGG analysis of shared significantly upregulated and downregulated genes between immature non-polarized neuronal growth stage (E12.5 v E17.5) and following AAV-Cited2-GFP overexpression (Cited2 v GFP). Source data are available online for this figure.

Dataset EV4). GO and KEGG pathway analysis of these shared genes revealed that upregulated transcripts showed an enrichment for terms involved in axon and dendrite growth, PNS development, and developmental processes (Fig. 5J; Dataset EV4), while the downregulated were associated with membrane and ion transport, synaptic regulation, lipid metabolism, and GPCR and kinase signaling (Fig. 5K; Dataset EV4) supporting our hypothesis that Cited2 overexpression promotes a more immature, non-polarized developmental state after a SCI.

As a validation of these findings, we confirmed an increase in expression of CBX5, involved in transcriptional and chromatin regulation (Bosch-Presegué et al, 2017) and HOXD3, associated with TGFb1/SMAD/BMP mediated signaling and involved in developmental processes including neural crest formation (Wang et al, 2020) after a DCA in AAV-Cited2 expressing DRG neurons (Appendix Fig. S4).

Moreover, CITED2 can influence chromatin accessibility via modulation of histone acetylation and recruitment of transcription factors (TFs) (Kranc et al, 2015; Li et al, 2012) (Appendix Table S1) associated with chromatin remodeling, including CBX5, TFAP2a and SMAD2/3 (Bamforth et al, 2001). Therefore, we performed ATAC-seq after Cited2 overexpression to examine whether Cited2-dependent changes in chromatin accessibility can reprogram adult sensory neurons towards an immature non-polarized maturation state after a SCI (Fig. 6A).

ATAC-seq, which reveals chromatin accessibility, allows identification of transcriptional programs that are associated with cellular states and identity. ATAC-seq analysis revealed 1436 and 2615 significantly differentially accessible promoter genes or enhancer regions (DA regions), respectively (Fig. 6B). Heatmap and hierarchical clustering of DA promoter or enhancer regions indicated different clustering between Cited2 overexpression or GFP control samples (Fig. 6C,D).

DAGs (Dataset EV5) that were more accessible included genes associated with development (e.g., Notch1, Neurod2, Cit, Cops2, Crim1, and Wnt9b), chromatin regulators (Dnmt3a and Kdm2b), TF regulators (Arx, Gbx, Klf1, Gal, Sirt3, Sirt5, Sox18, Trp53tg5, Trp53cor1, Foxc2, and Il6st), homeodomain genes (Rhox4e, Rhox4b, Rhox2d, Hoxb9, Hoxd13, and Zeb1), cell cycle (Terf2ip, Bab1, and Cdk5), translation (Eif3i and Rpl23a), cytoskeletal organization and axonal transport (Itga5 and Kif9), and neurite outgrowth (Zap70, Negr1, and Pgp). Functional analysis of DAGs indicated similar results; enriched for terms such as transcription, developmental processes, and Notch, Wnt, and GTPase signaling (Fig. 6E). Similarly, less accessible genes (Dataset EV5) were associated with transcriptional repressors (Zhx1) and synapse regulators (Syt4 and Syndig1), in line with GO and KEGG analysis

indicating an enrichment for terms associated with transcriptional repression, GPCR activity, and sensory perception (Fig. 6E).

Next, since enhancers are essential regulators of gene expression (Schoenfelder and Fraser, 2019), we correlated predicted promoter targets of enhancers from a previously published dataset (Shen et al, 2012) to our DA enhancer regions, as previously described (Palmisano et al, 2019) (Dataset EV5). Functional analysis of these predicted targets indicated more accessible regions were associated with transcription (Ctcfl, Dnmt1, Dnmt3a, Brd3, E2f2, Sox8, Stat6, and Smad3), cell cycle regulation (Bach2; Dab2ip; Ddit34), and protein binding, GTPase, MAPK, and Calcium/cAMP signaling (Dlk1, Bmp7, Smad3; Map3kl, Map4kl, Pak4, Cgnl1, Zabp1) (Fig. 6F). Less accessible regions were associated with ion transport and cell cycle (Cdkn2a and Mcc) (Fig. 6F). The above functional categories suggested that Cited2 might influence neuronal maturation, in line with RNA-seq results.

Additionally, while Cited2 itself does not have a DNA binding site (Bhattacharya et al, 1999), it was shown or predicted to directly and indirectly regulate the function of numerous TFs (Appendix Table S1), several of which are known to be involved in axonal growth and developmental processes, including TFAP2, CDX2, CEBPB, CREB1, CREBBP, EGR1, EP300, FOS, JUN, KLF, MEF2 SMAD, and STAT family members (Liu et al, 2011; Fawcett and Verhaagen, 2018), among others, to promote transcriptional and epigenetic changes. Given that Cited2 overexpression caused an enrichment in accessible promoter and enhancer regions, we investigated TF binding to accessible cis-regulatory elements via differential TF footprinting analysis.

We first applied Bivariate Genomic Footprinting (BaGfoot) analysis (Baek et al, 2017) to identify putative transcription factors (TFs) bound to, or released from, chromatin regions exhibiting changes in accessibility. This approach measures the change in accessibility within a TF motif, providing information about footprint depth, an indicator of TF binding. Bound TFs typically display a deeper footprint depth (FDP; y-axis), while regions of increased chromatin accessibility exhibit greater flanking accessibility (FA; x-axis). Four quadrants are thereby defined: quadrant I represents TFs bound at more accessible regions; quadrant II, TFs bound at less accessible regions; quadrant III, TFs released from less accessible regions; and quadrant IV, TFs released from more accessible regions.

At the promoter level (Dataset EV6; Data S7), our analysis revealed enhanced chromatin accessibility through recruitment of TFs such as TFAP2A, CDX1, NEUROG2, Homeobox, and STAT family members (Fig. 7A, quadrant I), accompanied by release of TFs such as TP53, MYB, MYC, and POU-domain proteins (Fig. 7A, quadrant IV). No TFs were associated with reduced chromatin

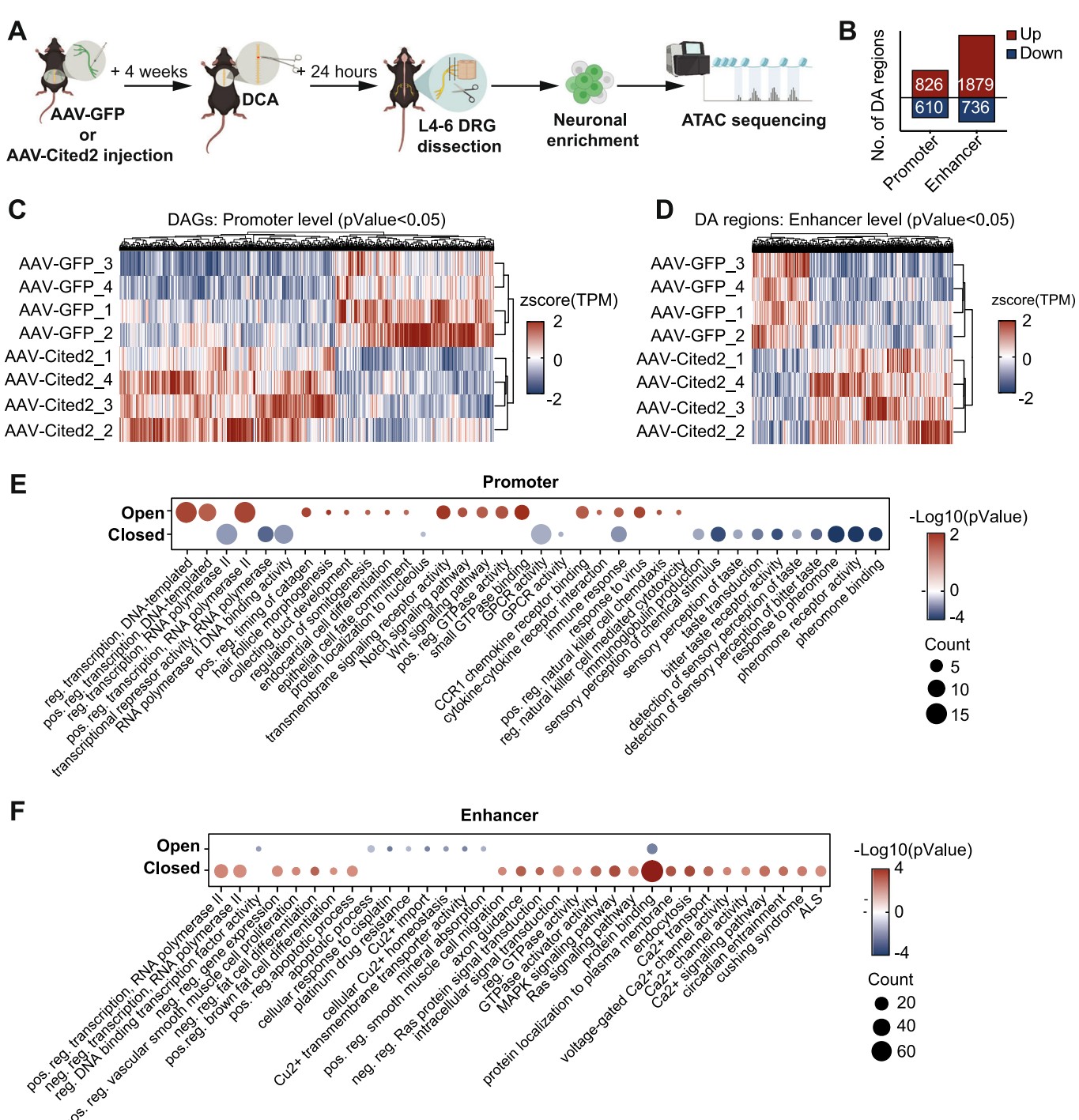

**Figure 6. Cited2 overexpression after SCI promotes increased differential accessibility of promoter and enhancer regions that is associated with developmentally regulated genes.**

(A) Experimental design (made with BioRender), $n = 4$ independent biological replicates were input for ATAC-seq. (B) Number of significantly differentially accessible regions (DA) at the promoter and enhancer (DAGs are identified using EdgeR, where the significance threshold is defined as $p$ value <0.05). (C, D) Heatmap and hierarchical clustering of DA promoter or enhancer regions from AAV-GFP control and AAV-Cited2-GFP samples. (E) GO and KEGG analysis of DA promoter regions. (F) GO and KEGG analysis of the predicted gene targets associated with DA enhancer regions. $P$ values are calculated using a modified Fisher's exact test as described in DAVID (Huang et al, 2009). Source data are available online for this figure.

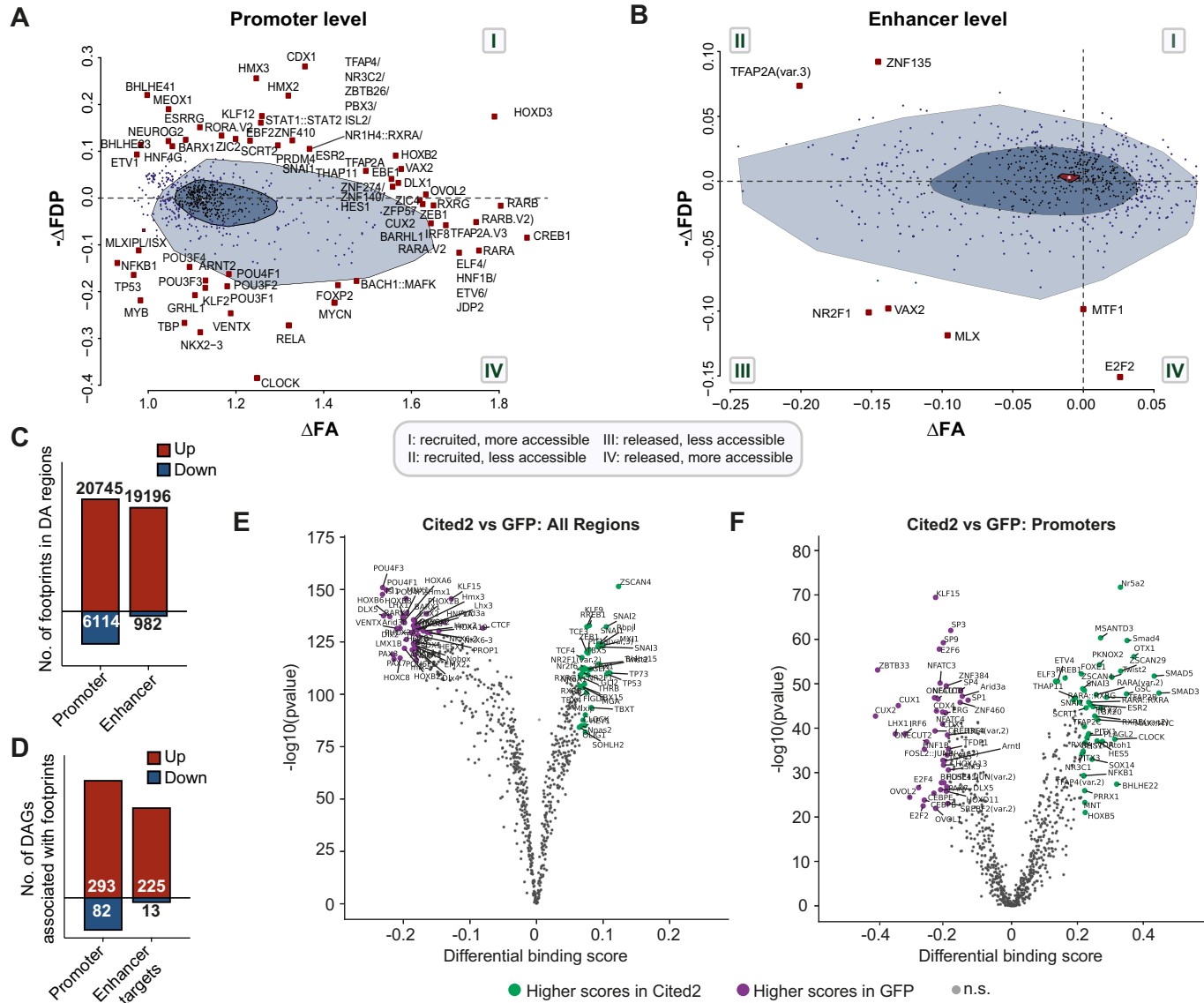

**Figure 7. Differential TF activity analysis suggests Cited2 overexpression promotes TF recruitment and release at accessible chromatin.**

$n = 4$ independent biological replicates were input for ATACseq. (**A, B**) BaGfoot plots showing differential flanking accessibility (ΔFA) and footprint depth (ΔFDP) of motifs after Cited2 v GFP at (**A**) promoters [transcription start site (TSS) ± 1 kb] and (**B**) enhancers. (**C**) Number of motif footprints derived from HINT analysis in DA promoter or enhancer regions ($p < 0.05$) ± 1.5 kb. (**D**) Number of DA promoter or predicted enhancer target genes associated with HINT-derived footprint motifs ($p < 0.05$). (**E, F**) Differential footprinting analysis of TF activity across all regions and DA promoter regions after Cited2 overexpression or GFP control, calculated using TOBIAS BINDetect (Bentsen et al, 2020). Significance is determined by comparing observed footprint $\log_2$ fold changes to background distributions, where $p$-values are estimated via subsampling. Source data are available online for this figure.

accessibility in flanking regions (Fig. 7A, absence of quadrants II and III).

At the enhancer level, fewer TFs were identified (Data S7; Fig. 7B), with enrichment for repressor recruitment, such as TFAP2A and ZNF135 (Fig. 7B, quadrant II), and release of activators including NR2F1, MLX, and VAX2 (Fig. 7B, quadrant III). Although fewer TFs were detected compared to promoter regions, the range of accessibility (x-axis) spanned nearly twice that observed following peripheral sciatic nerve injury (Palmisano et al, 2019), suggesting that CITED2 acts at both promoter and enhancer regulatory levels. Moreover, while promoter accessibility

changes were predominantly concordant (increased accessibility), enhancer effects were more variable, showing both increased and decreased accessibility.

To further identify active TF binding sites associated with CITED2 overexpression versus GFP control, we performed HMM-based Identification of Transcription factor footprints (HINT) analysis. This method detects TF footprint motifs in regions of accessible chromatin using ATAC-seq data, rather than relying on in silico predictions. At the promoter level, we identified 20,745 and 6114 TF footprint motifs in more and less accessible regions, respectively (Fig. 7C). At enhancers, 19,196 and 982 footprint

motifs were detected in more and less accessible regions, respectively (Fig. 7C). Differentially accessible promoter footprints were associated with 376 genes, while enhancer footprints corresponded to 239 predicted target genes (Fig. 7D). Together, these results are consistent with the BaGfoot analysis, supporting a model in which CITED2 overexpression promotes a more accessible chromatin conformation.

Second, we used TOBIAS (Bentsen et al, 2020) to perform differential TF analysis across all regulatory regions and at differentially accessible regions to identify changes in TF activity associated specifically with either Cited2 overexpression or GFP control (Fig. 7E,F; Dataset EV6). Results indicated that Cited2 overexpression increased activity of TFs, such as Basic-helix-loop-helix (bHLH) (e.g., Bhlh22, Twist1, and Atoh1), TFAP2, TFAP5, Homeobox factors (e.g., HoxB5), as well as TFs associated with cell fate decisions (e.g., SNAIs), which co-exist with activators found via BaGfoot analysis that led to increased chromatin accessibility, as well as to CITED2 predicted protein interactors (Appendix Table S1) such as Smad3, Smad5, TP73, and ETV4. On the other hand, TF activity associated with the GFP control condition reflected activity of transcriptional repressors, such as PAX, POU, and ONECUT factors, which were previously found to be recruited to flanking regions in DRG following non-regenerative spinal injuries (Palmisano et al, 2019).

Together, these data provide evidence that Cited2 overexpression enhances chromatin accessibility and triggers the enrichment of transcriptional activation associated with neuronal developmental and immature growth states.

We then examined the correlation between gene expression and ATAC signal at promoters following Cited2 overexpression after a SCI. Firstly, ATAC signal density plots indicated that upregulated genes showed increased ATAC signal near the transcription start site (TSS) following Cited2 overexpression compared to down-regulated genes (Appendix Fig. S5).

Next, we explored the function of genes with increased expression and promoter accessibility following Cited2 over-expression. Unbiased correlation analysis of normalized gene expression and chromatin accessibility at the promoter indicated a moderate correlation [$R^2 = 0.34$ (Spearman); $p$ value <3.3e−16] (Fig. EV4A). Genes were organized into four groups to examine the relationship between gene expression and accessibility: genes were highly expressed and highly accessible (HE-HA; 7582 genes) if their normalized transcripts per million (TPM) was higher than the 75th percentile of the data and genes were low-to-medium expression and low-to-medium accessibility (ME-MA; 6420 genes) if their TPM was lower than the 50th percentile of the data (Fig. EV4A; Dataset EV7). Functional analysis of HE-HA genes indicated terms associated with transcription, nervous system development and neurogenesis, protein and metabolic processes, cell cycle/apoptosis, transport, cell signaling and synaptic regulation, while ME-MA genes were associated with immune response and GPCR signaling (Fig. EV4B; Dataset EV7). Similar functional categories were observed for differentially expressed or accessible genes alone, as shown previously (Figs. 5E and 6G,H). Additionally, odds ratio and Fisher's test analysis indicated a significant correlation between differentially expressed genes and high promoter accessibility, but not low-to-medium promoter accessibility (Fig. EV4C). Similar analysis of differentially accessible and expressed genes revealed a significant correlation specifically

between upregulated genes and increased chromatin accessibility (Fig. EV4D) indicating that gene expression and epigenetic remodeling may be achieved through TF activation following Cited2 overexpression to promote a more immature, non-polarized neuronal growth state after a SCI.

Together, gene expression signatures associated with a neuronal immature state correlated to more accessible chromatin following Cited2 overexpression and SCI.

Finally, in order to assess whether Cited2 expression would affect neuronal maturation, cultured embryonic DRG neurons were transduced with either AAV-Cited2-GFP or control AAV-GFP, and DRG neuron maturation was scored as previously detailed (Nascimento et al, 2022), reflecting the degree of neuronal morphological changes (from multipolar to bipolar, bell-shaped bipolar and pseudo-unipolar morphologies). Strikingly, Cited2 overexpression significantly regulated neuronal maturation, resulting in the majority of DRG neurons with a multipolar or bipolar morphology and reducing the percentage of pseudo-unipolar neurons (Fig. 8A,B). Cited2 overexpression in cultured embryonic DRG neurons also promoted the upregulation of Cbx5 and HoxD3 (Fig. 8C,D), consistent with the in vivo results after a SCI.

Together, our data indicate that the loss of growth potential is marked by epigenetic and transcriptional changes that can be reversed by manipulation of Cited2, which was sufficient to shift chromatin to a more accessible growth-competent state after SCI.

## Panobinostat enhances DRG neuronal growth via a Cited2-dependent mechanism

Next, we sought to identify a small-molecule compound capable of modulating CITED2 expression to promote repair after SCI. Using the Ingenuity Pathway Analysis (IPA) database, we screened published and predicted compounds with the potential to enhance Cited2 transcription. This analysis identified Panobinostat, a pan-histone deacetylase (HDAC) inhibitor (Laubach et al, 2021) (Appendix Fig. S6A), as the only compound predicted to selectively upregulate Cited2 expression while concomitantly increasing histone acetylation.

We first tested whether administration of Panobinostat would promote DRG neurite outgrowth. Indeed, we found a significant dose-dependent increase in neurite outgrowth, with 200 nM of Panobinostat exerting the maximal effect (Appendix Fig. S6B,C). We next confirmed that Panobinostat enhanced the expression of Cited2 seen by immunofluorescence (Appendix Fig. S6D,E) and RT-qPCR (Appendix Fig. S6F). Further, Cut and Tag for H3K27ac on Cited2 promoter regions from NeuN expressing DRG neurons revealed that Panobinostat increased Cited2 promoter acetylation (Fig. EV5), in line with increased expression. Finally, we examined whether Panobinostat-induced neurite outgrowth was dependent on Cited2. Data analysis indicated that indeed CRISPR/Cas9-mediated Cited2 knockdown significantly reduced Panobinostat-dependent outgrowth in DRG sensory neurons (Appendix Fig. S6G,H), showing dependence of growth-promoting effects of Panobinostat on Cited2. Additionally, we found that Panobinostat delivery in cultured DRG neurons promoted increased nuclear expression of p300/CBP (Appendix Fig. S6I,J), and augmented acetylation of H3K9 and H3K27 (Appendix Fig. S6K–N), in line with results following Cited2 overexpression.

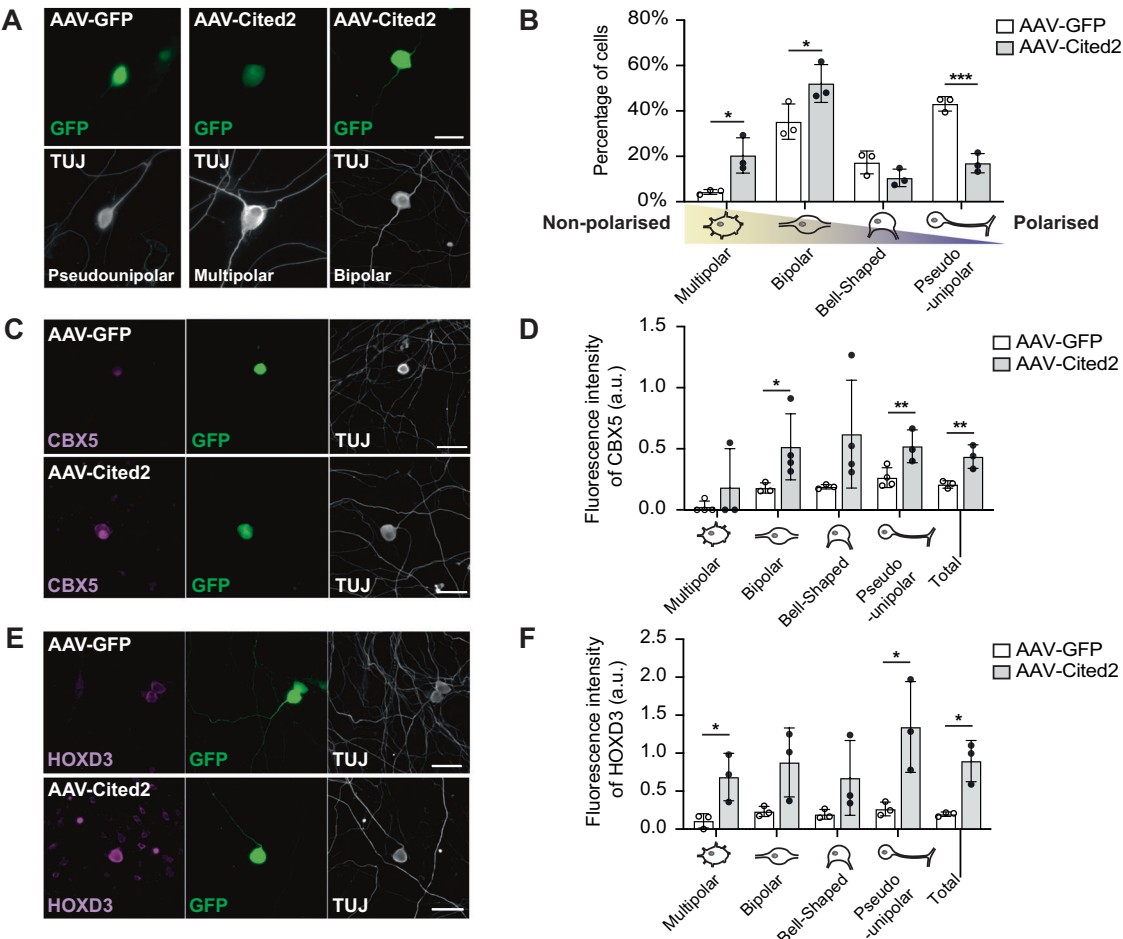

**Figure 8. Cited2 overexpression in DRG neurons affects maturation.**

(A) DRG cultures (E13.5) transduced with AAV-GFP or AAV-Cited2-GFP (green), showing changes in maturation (white: TUJ). (B) Percentage of cells in different DRG maturation stages (two-way ANOVA with Tukey's post hoc test; Group: $f(1) = 1.15e-18$, $p = 1$; Morphology: $f(3) = 40.51$, $p = 1.061 \times 10^{-7}$); Interaction score: $f(3) = 19.60$, $p = 1.315 \times 10^{-5}$; $n = 3$ independent biological replicates, at least 50 cells per replicate measured. Displayed are $p$ values from Tukey's post hoc test: multipolar: $p = 0.0494$, bipolar: $p = 0.0376$, bell-shaped: $p = 0.8161$, pseudo-unipolar: $p = 0.0008$. (C, D) CBX5 immunostaining (magenta) in DRG neurons (TUJ, white) after AAV-GFP or AAV-Cited2-GFP overexpression (green) in E13 DRG culture with quantification of fluorescence intensity by morphology and overall (Multipolar: $p = 0.1995$, bipolar: $p = 0.0263$, Bell-shaped $p = 0.0623$, pseudo-unipolar: $p = 0.0037$, Total: $p = 0.0015$; $n = 3$-4 independent biological replicates, at least 50 cells per replicate measured). (E, F) HOXD3 immunostaining (magenta) in DRG neurons (Tuj, white) after AAV-GFP or AAV-Cited2-GFP overexpression (green) in E13 DRG culture with quantification of fluorescence intensity (Multipolar $p = 0.0375$; bipolar: $p = 0.0715$; bell-shaped: $p = 0.1695$; pseudo-unipolar: $p = 0.0365$; total: $p = 0.0113$; $n = 3$ independent biological replicates, at least 50 cells per replicate measured. Analysis by individual $t$-tests). Scale bars: 50 µm. *$p < 0.05$, **$p < 0.01$, ***$p < 0.001$, and ****$p < 0.0001$. ns not significant. All error bars shown as standard deviation (SD). Source data are available online for this figure.

## Panobinostat promotes axonal growth, sprouting and neurological recovery after SCI

To evaluate whether the growth-promoting effects of Panobinostat could be translated in vivo, we administered the compound systemically via daily intraperitoneal (i.p.) injections at two clinically relevant doses (Laubach et al, 2021) (15 and 20 mg/kg) over a 5-day period. Ex vivo quantification of regenerative growth revealed a significant increase in average DRG neurite outgrowth following treatment with 20 mg/kg Panobinostat (Appendix Fig. S7A,B). Immunoblot analysis of ex vivo–isolated DRGs showed that this enhanced outgrowth was accompanied by elevated levels of H3K9ac and H3K27ac (Appendix Fig. S7C). Next, we investigated whether Panobinostat could promote the

growth of central sensory axons following spinal cord injury (SCI). A T9 dorsal column hemisection was performed, sparing tissue ventral to the central canal but resulting in hindlimb paralysis. Approximately 3 h post-injury, animals received either vehicle or 20 mg/kg Panobinostat via i.p. injection. Treatment followed a 5-day-on/2-day-off regimen for 6 weeks. At week 5, Dextran-594 was bilaterally injected into the sciatic nerves to label ascending sensory axons within the dorsal columns (Fig. 9A).

Treatment with Panobinostat significantly enhanced axonal growth into and beyond the lesion site compared with vehicle-treated controls (Fig. 9B, C). Notably, GFAP expression within the lesion area was unchanged between groups, indicating that Panobinostat did not affect astroglial reactivity (Fig. 9D).

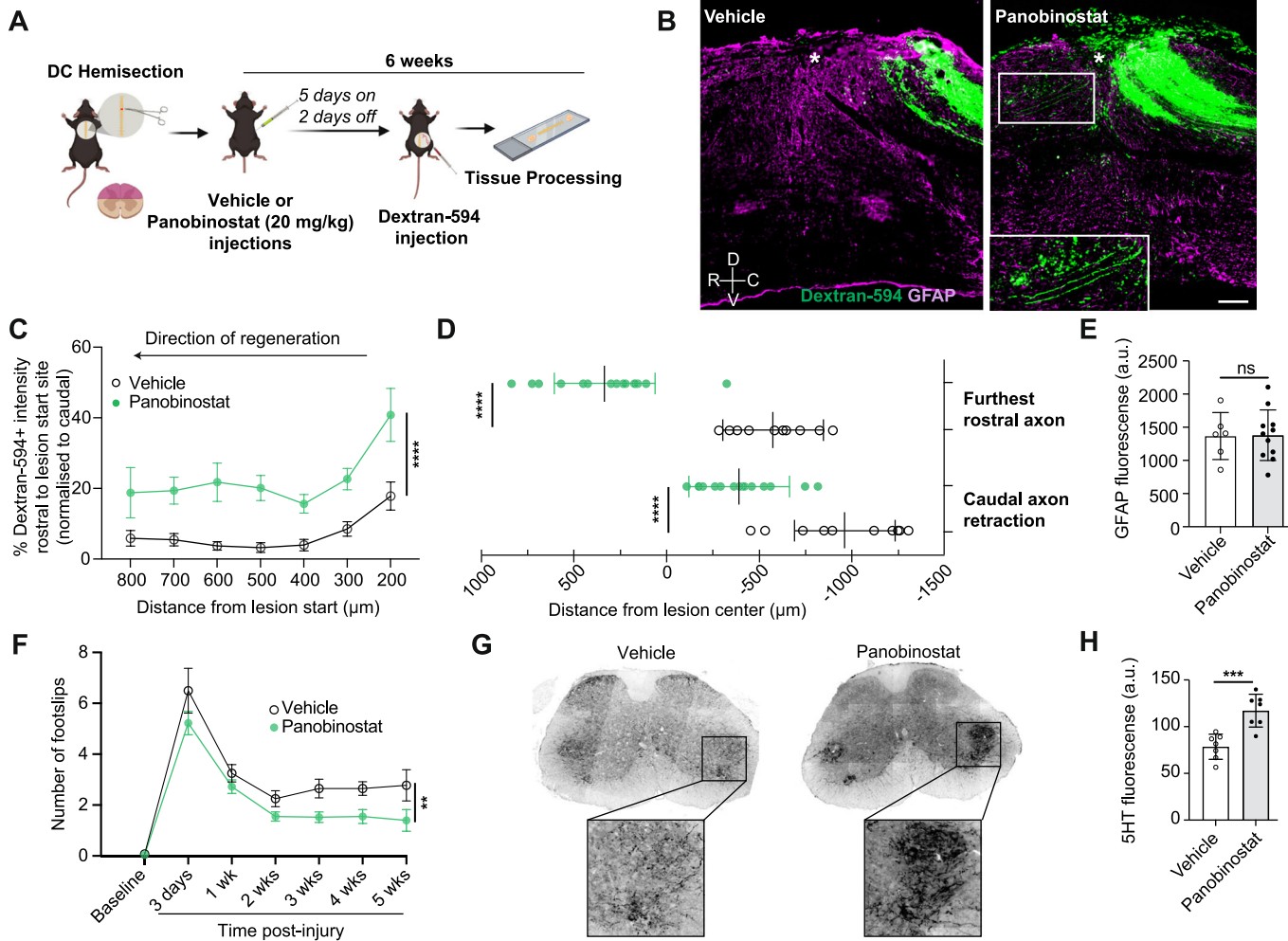

**Figure 9. Panobinostat promotes axonal growth and increases serotonergic sprouting after a spinal dorsal column overhemisection.**

(A) Experimental design (made with BioRender). (B) Representative micrographs of vehicle or 20 mg/kg Panobinostat i.p. delivery six weeks after a dorsal column overhemisection. Dextran-594 (green) is used to trace ascending dorsal column tracts. GFAP (magenta) is used to determine the lesion epicenter (white star). Scale bar: 200 μm. (C) Percentage Dextran-594 intensity rostral to the lesion border in vehicle or Panobinostat-treated mice after SCI. Fluorescence intensity rostral to the lesion border is normalized to Dextran-594 intensity caudal to the lesion site. Mixed-effects ANOVA with uncorrected Fisher's LSD post hoc test; Vehicle: $n = 11$ biological replicates, Panobinostat: $n = 12$ biological replicates. Treatment: $f(1,21) = 32.70$, $p < 0.0001$; Distance: $f(2.21,46.05) = 5.97$, $p = 0.0038$); Interaction score: $f(6,125) = 0.52$, $p = 0.7920$. Displayed is the treatment $p$ value. (D) Average distance between the lesion epicenter and the furthest rostral axon ($t$-test, $p = 1.579 \times 10^{-8}$) or retracted caudal axons ($t$-test, $p = 1.410 \times 10^{-5}$). Vehicle: $n = 10$, Panobinostat: $n = 15$ biological replicates. (E) Quantification of average GFAP intensity in vehicle and Panobinostat-treated mice ($t$-test, $p = 0.9487$). Vehicle: $n = 6$, Panobinostat: $n = 11$ biological replicates. (F) Gridwalk quantification of the average number of missteps per run in Vehicle or Panobinostat-treated mice after SCI. Two-way ANOVA with Sidak post hoc test; $n = 10$ biological replicates. Treatment: $f(1,9) = 12.71$, $p = 0.0049$; Time: $f(2.23, 20.07) = 44.21$, $p < 0.0001$; Interaction score: $f(2.39, 21.56) = 0.91$, $p = 0.4329$. Displayed is the treatment $p$ value. (G, H) Representative micrographs and quantification of serotonergic (5-HT$^+$ axons (black) in the ventral horn of the lumbar enlargement caudal to the lesion site ($t$-test, $p = 0.0006$, $n = 7$ biological replicates). Scale bar: 200 μm. *$p < 0.05$, **$p < 0.01$, ***$p < 0.001$, and ****$p < 0.0001$. ns not significant. All error bars shown as standard deviation (SD). Source data are available online for this figure.

To determine whether systemic Panobinostat administration also improved neurological recovery, we performed the gridwalk test, quantifying hindlimb foot slips during successive runs across a wire grid. Panobinostat-treated mice exhibited a significant reduction in foot slips by week three post-injury, indicating improved motor coordination (Fig. 9E).

Functional recovery could result from the growth and sprouting of descending motor projections that partially compensate for lost connections. Among these, the raphespinal (serotonergic; 5-HT) tract provides critical input to spinal motor circuits and

may serve as a compensatory "detour" pathway. Immunohistochemical analysis revealed that Panobinostat treatment significantly increased the density of 5-HT-positive fibers within the ventral horn of the lumbar enlargement, particularly in laminae containing motoneurons controlling hindlimb movement (Fig. 9F,G).

Collectively, these findings indicate that Panobinostat enhances both central axon regeneration and 5-HT circuit plasticity, contributing to improved functional recovery following spinal cord injury.

# Discussion

Our findings provide direct experimental evidence that neuronal maturation is inversely correlated with the regenerative ability of DRG sensory neurons and the loss of regenerative competence is an epigenetically regulated process. Importantly, the development of human cortical neurons is regulated by an intrinsic cellular epigenetic clock that enables neuronal maturation (Ciceri et al, 2024), lending human relevance to our data. Neuronal maturation seems to restrict the access of essential transcriptional proteins to developmentally repressed chromatin regions, thus failing to induce the expression of genes that are essential for axon regeneration (Grewal and Moazed, 2003; Brown et al, 1997; Peters et al, 2001). Whether this developmental progression is permanent or it can be reversed epigenetically had not been directly addressed so far. Importantly, here we found that loss of epigenetic-dependent growth competence is reversible by modulating the epigenetic regulator CITED2. We identified CITED2 as an epigenetic cofactor whose expression was increased in DRG after a regenerative competent SNA but not after a regenerative incompetent DCA. Previous studies have shown that CITED2 is extensively involved in the regulation of developmental processes. These include neural tube closure, the development of the lens, heart, DRG, neocortical and callosal projection neurons (Kranc et al, 2015; Bamforth et al, 2001; Li et al, 2012; Weninger et al, 2005; Chen et al, 2008; Fame et al, 2016; Chen et al, 2009; Sun et al, 2007) as well as in embryonic and adult stem cell proliferation and differentiation (Kranc et al, 2009; Withington et al, 2006; Li et al, 2012).

CITED2 overexpression was sufficient to induce transcriptional and chromatin accessibility signatures associated with immature neuronal states, steering neurons towards an immature growth-competent state. Specifically, CITED2 overexpression induced increased chromatin accessibility, increased expression of EP300 and active CBP expression, and increased acetylation levels of H3K9 and H3K27 after SCI, which have previously been linked with an enhanced regenerative state (Palmisano et al, 2019; Hervera et al, 2019; Hutson et al, 2019; Puttagunta et al, 2014; Gaub et al, 2011; Ewan et al, 2021; Finelli et al, 2013). This resulted in upregulation of genes required for axon and dendrite growth, PNS development and downregulation of genes specific for neuronal maturation and synaptic activity. This epigenetic and transcriptional reprogramming was sufficient to promote axon growth after a SCI, indicating that CITED2 reprograms mature DRG neurons to re-activate the growth potential that they lost during maturation, through epigenetic modifications. CITED2-dependent enhancement in histone acetylation, which is sustained for weeks after SCI, facilitated the recruitment and activation of transcription factors at open chromatin regions. Our TF footprint analysis revealed several transcriptional regulators that were preferentially recruited at chromatin regions made accessible by CITED2 overexpression. Specifically, we identified factors belonging to the basic-helix-loop-helix and HOX families, key regulators of nervous system development (Miller and Dasen, 2024; Tutukova et al, 2021).

Among them, ATHO1 has been implicated in neurogenesis in the retina (Todd et al, 2021); MEIS1 has been implicated in eye and tail regenerative in planaria (Wang et al, 2023); MEIS2 was identified as a putative transcriptional regulator of genes associated to retinal ganglion cell differentiation and neuron development in regenerating conditions (Jacobi et al, 2022); FOXG1 has been

shown to protect neurons from apoptosis (Dastidar et al, 2011); NKX2-5 has been involved in myocardial repair (De Sena-Tomás et al, 2022).

It is also possible that increased levels of H3K27ac and H3K9ac are driven by an increase in CITED2-mediated activation of histone acetyltransferases (HATs), p300 and CBP, as well as their associated factor, P/CAF. These HATs function to coordinate and integrate multiple upstream signaling events in developmental biological processes and disease states, through their ability to recruit a variety of TFs to genomic sites that are hyperacetylated, including after neuronal injury, where they can promote axon growth (Hezroni et al, 2011; Zhong and Jin, 2009; Palmisano et al, 2019; Hutson et al, 2019; Puttagunta et al, 2014; Gaub et al, 2011).

Following CITED2 overexpression, the DRG neurons do not seem to lose their identity or regress to embryonic stages that precede their neuronal specification. Therefore, this seems different from a report describing the interesting findings that corticospinal neurons express gene signatures associated with early embryonic states after a SCI (Poplawski et al, 2020), including boosting growth in association with spinal cord stem cell transplants. As an alternative molecular approach to ours, reducing DNA methylation has been shown to counteract age-dependent regenerative decline of retinal ganglia neurons after optic nerve injury (Lu et al, 2020). Our data are in accordance with previous studies, where dynamic gene expression changes in mouse DRG neurons between the neuronal growth stage E12.5 and a peripheral injury were observed. These reports have found that silencing synaptic proteins such as the voltage-gated calcium channel subunit *a2d2*, or vesicle priming proteins at the presynaptic active zone, can promote axon growth after a SCI (Tedeschi et al, 2016; Hilton et al, 2022). However, whether these mechanisms relied on epigenetic mechanisms and whether those interventions led to a reversal of these neurons to a presynaptic state remains elusive.

By screening for putative pharmacological compounds that could enhance Cited2 expression and histone acetylation after SCI, we identified the HDAC-inhibitor Panobinostat, which we increased Cited2 expression and promoter acetylation in DRG neurons.

Panobinostat targets class I and class II HDACs, leading to enhanced histone acetylation and gene transcription. Despite this, we found that Panobinostat-dependent neuronal growth required CITED2, given that Cited2 KD in the presence of Panobinostat abrogated the gain in neuronal outgrowth. The increase in Cited2 expression and promoter histone acetylation after Panobinostat is in line with the evidence that the Cited2 promoter is acetylated during DRG neuronal maturation and following a regenerative sciatic nerve injury. Cited2 is therefore responsive to increased histone acetylation driven by Panobinostat-dependent HDAC-inhibition. Importantly, Panobinostat enhanced sensory axon growth, 5-HT sprouting and neurological recovery after SCI. While other broad HDAC inhibitors have been reported to promote neuroprotection and repair after SCI (Finelli et al, 2013), here Panobinostat significantly enhanced axonal growth and sprouting. Whether the long-term administration of Panobinostat to enhance CITED2 expression and promote neuronal regeneration may potentially affect neural structural stability remains unknown. However, notably, Panobinostat is already approved for oral administration as part of combination therapy in multiple myeloma, where it exhibits favorable tolerability and robust efficacy

even at low doses (Laubach et al, 2021; Pan et al, 2023). It will be interesting to investigate whether combining Panobinostat with neurorehabilitation, biomaterials, stem cell grafts, or fibrotic scar-reducing compounds, such as those known to digest chondroitinase, can further boost axon growth, repair and recovery after SCI. In summary, the main implications of these findings lie (i) in the discovery of a mechanism that can be leveraged to reprogram neuronal chromatin toward a growth-competent state, thereby restoring the regenerative capacity of adult sensory neurons after SCI; (ii) in the identification of a druggable target with a clinically approved compound, which promotes repair and recovery after SCI.

# Methods

## Reagents and tools table

| Reagent/resource | Reference or source | Identifier or catalog number |
|---|---|---|
| **Experimental models** | | |
| C57BL/6 (*M. musculus*) | Charles River Laboratories | C57BL/6NCrl |
| F11 cells | Sigma | 08062601 |
| **Recombinant DNA** | | |
| AAV8-CAG-WPRE-EGFP-p2A-Cited2 | Duke University: Viral Vector Core | pBK1187 |
| AAV8-CAG-WPRE-EGFP | Duke University: Viral Vector Core | pBK313 |
| CRISPR-Case9 Cited2 double nickase plasmid | Santa Cruz | sc-400246-NIC |
| pcDNA3.1GFP | Addgene | 70219 |
| pcDNA3.1Cited2 | Addgene | 48184 |
| **Antibodies** | | |
| Mouse anti-Cited2 | Santa Cruz | Sc-21795 |
| Rabbit anti-H3K27ac | Abcam | Ab4729 |
| Rabbit anti-H3K9ac | Abcam | Ab10812 |
| Chicken anti-GFP | Abcam | 13970 |
| Rabbit anti-GFAP | Millipore | AB5804 |
| Chicken anti-NeuN | Millipore | ABN91 |
| Chicken anti-Beta 3 tubulin (Tuj) | Abcam | 18207 |
| Rabbit anti-beta-actin | Abcam | 155279 |
| Rabbit anti-Gapdh | Cell Signaling | 2118S |
| Rabbit anti-HP1a (Cbx5) | Cell Signaling | 2616BC |
| Rabbit anti-HoxD3 | Sigma-Aldrich | AV31964 |
| Rabbit anti-Sox10 | Abcam | 155279 |
| Rabbit anti-p300 | Cell Signaling | 86377S |
| Rabbit anti-acCBP | Abcam | Ab61242 |
| Mouse anti-NF200 | Sigma | N0142 |

| Reagent/resource | Reference or source | Identifier or catalog number |
|---|---|---|
| Rabbit anti-H3k27ac | Abcam | Ab177178 |
| Guinea pig anti-rabbit IgG | Antibodies Online | ABIN101961 |
| **Oligonucleotides and other sequence-based reagents** | | |
| CUT&Tag primers | This study | Sequence available on request |
| **Chemicals, enzymes and other reagents** | | |
| Dispase II | Sigma | D4693 |
| Collagenase Type II | Gibco | 17504-044 |
| Papain papaya latex | Sigma | P3125 |
| SUPERase In RNase Inhibitor | Invitrogen | AM2696 |
| cOmplete Protease Inhibitor Cocktail | Roche | 11697498001 |
| Phosphatase inhibitor | Roche | 4906845001 |
| RNeasy Mini Kit | Qiagen | 74104 |
| SuperScript First-Strand Synthesis System Kit | Thermo Fisher Scientific | 18091050 |
| HotStart ReadyMix | Kapa | 07958935001 |
| Dextran-594 | Thermo Fisher Scientific | D22913 |
| Panobinostat | Selleckchem, UK | LBH589 |
| RNeasy Micro Kit | Qiagen | 74004 |
| TruSeq Stranded mRNA kit | Illumina | RS-122-2101 |
| Concanavalin A-coated magnetic beads | Bangs Laboratories | BP531 |
| CUTANA pAG-Tn5 | Epicypher | 15-1017 |
| Zymo DNA Clean & Concentrator kit | Zymo Research | D4013 |
| **Software** | | |
| Fiji (ImageJ) | Schindelin et al, 2012 | |
| Cytoscape-v3.9.1 | Shannon et al, 2003 https://cytoscape.org | |
| BioVenn | Hulsen et al, 2008 | |
| Adobe Illustrator | https://www.adobe.com | |
| BioRender | BioRender.com | |
| Bcl2fastq-v2.20 | Illumina | |
| FastQC-v0.11.9 | Andrews, 2010 https://www.bioinformatics.babraham.ac.uk/projects/fastqc/ | |
| TrimGalore!-v0.6.6 | Krueger, 2012 https://github.com/FelixKrueger/TrimGalore | |
| Salmon-v1.6.0 | Patro et al, 2017 https://github.com/COMBINE-lab/salmon | |
| DESeq2-v1.34.0 | Love et al, 2014 https://bioconductor.org/packages/release/bioc/html/DESeq2.html | |

| Reagent/resource | Reference or source | Identifier or catalog number |
|---|---|---|
| SRAToolkit-v2.11.2 | NIH https://hpc.nih.gov/apps/sratoolkit.html | |
| GeneOverlap | Shen, 2022 https://github.com/shenlab-sinai/GeneOverlap | |
| Kundaje lab's ATAC-seq processing pipeline | https://github.com/kundajelab/atac_dnase_pipelines | |
| HTseq-v.0.6.1 | https://htseq.readthedocs.io/en/latest/ | |
| EdgeR-v3.8.6 | https://bioconductor.org/packages/release/bioc/html/edgeR.html | |
| GenomicAlignment-v1.32.0 | https://www.bioconductor.org/packages/release/bioc/html/GenomicAlignments.html | |
| NGSplot-v2.47.1 | https://github.com/shenlab-sinai/ngsplot | |
| DAVID-v2012 | Sherman et al, 202, Huang et al, 2009 https://davidbioinformatics.nih.gov | |
| ClueGo | Bindea et al, 2009 (Cytoscape App) https://apps.cytoscape.org/apps/cluego | |
| Graphpad Prism-v.9.4.0 | https://www.graphpad.com | |
| Ggplot2-v.3.3.6 | Wickham, 2016 https://ggplot2.tidyverse.org | |
| HINT-v0.13.2 | Li et al, 2019 https://reg-gen.readthedocs.io/en/hint/introduction.html | |
| BaGfoot | Baek et al, 2017 | |
| TOBIAS | Bentsen et al, 2020 https://github.com/loosolab/TOBIAS | |
| Pscan | Wickham et al, 2009 http://159.149.160.88/pscan/ | |
| **Other** | | |
| DNA lo-bind Eppendorf | Eppendorf | EP0030108051 |
| Applied Biosystems 7900HT FAST | Thermo Fisher Scientific | |
| 2100 Bioanalyzer | Agilent | |
| HiSeq 4000 | Ilumina | |

## Mice

All experimental procedures were performed on 6–8-week-old, male, wild-type C57BL/6 mice (Charles River) and approved by Imperial College Ethics Committee, in accordance with the UK Animals (Scientific Procedures) Act, 1986. Animals were housed in regular light/dark cycles with food and water ad libitum. Prior to surgeries, mice received a subcutaneous injection of analgesics (Rimadyl, 5 mg/kg and Buprenorphine, 0.1 mg/kg), and were anaesthetized with Isoflurane (4% induction, 2% maintenance, 1% O2). Surgery was performed on a heated pad to maintain body temperature at 37 °C.

## Injury models

For SNA, the hind limbs were shaved and incised ~2 mm from the L4–L6 DRG Muscles were displaced, the sciatic nerve exposed and axotomized using iridectomy scissors (FST). The nerve was then placed back, and the skin was fully closed with suture clips. For sham surgeries, the same procedure was performed except that the nerve was left untouched. Mice were killed 24 h after SNA for tissue processing. For laminectomy or DCA, animals were shaved around the thoracic area. A dorsal skin incision was made, and the superficial tissue was separated. Muscle tissue was opened to expose T9-11. A T9 laminectomy was performed to expose the T9 spinal cord, and then a DCA was carried out at T9 using iridectomy scissors to excise the dorsal columns bilaterally. The muscles surrounding the injury site were sutured, and the skin was closed using clips. For LAM control, the above procedure was followed, but no dorsal column injury was performed. Mice were killed 24 h or 6 weeks after DCA for tissue processing.

## AAV administration

Control AAV8-CAG-WPRE-EGFP (AAV-GFP) or AAV8-CAG-WPRE-EGFP-p2A-Cited2 (AAV-Cited2-GFP) was delivered into the sciatic nerve 4 weeks prior to a DCA. Briefly, the sciatic nerve was exposed as described previously and closed Dumont forceps (FST) were placed underneath to provide support. A Hamilton syringe was used to inject 3 µl of AAV-GFP or AAV-Cited2-GFP bilaterally into the lumen of the sciatic nerve over 5 min. Following injections, the nerve was repositioned, and the skin was closed using suture clips.

## Dorsal column tracing

Five weeks after a dorsal column overhemisection, Dextran-594 (15% Dextran Alexa Fluor 594, 10000 MW, Thermo Fisher Scientific) was delivered into the sciatic nerve. Briefly, the sciatic nerve was exposed as described previously and closed Dumont forceps (FST) were placed underneath to provide support. A Hamilton syringe was used to inject 3 µl of Dextran-594 bilaterally into the lumen of the sciatic nerve over 5 min to label ascending sensory axons within the dorsal columns. The syringe was left in the nerve for three minutes after Dextran-594 was dispensed to ensure no backflow of the tracer. Following injections, the nerve was repositioned, and the skin was closed using suture clips.

## Panobinostat administration

Vehicle control (dimethyl sulfoxide; DMSO) or 20 mg/kg of Panobinostat (Selleckchem, UK) was systemically delivered through i.p. injections two hours after injury on a 5-day on, 2 days off basis for a total period of 6 weeks. The last injection was given 2 h before sacrifice.

## Tissue dissection

For DRG dissection, the mouse skin, head, forelimbs, and organs were removed. The mouse was placed dorsal side down; muscles and ligaments were carefully removed to expose the vertebra. The vertebra was then cut using spring scissors (FST) and removed to expose the spinal cord and DRG. The sciatic nerve was identified

and followed to identify L4-6 DRG, which were removed and placed in an appropriate solution depending on use. For spinal cord dissection, mice were perfused transcardially with ice-cold 1x PBS followed by 4% paraformaldehyde (PFA). The mouse was prepared as described above, and care was taken to isolate and remove the spinal cord and DRG from surrounding tissue.

## Immunohistochemistry

Dissected tissue was post-fixed in 4% PFA and transferred to 30% sucrose for cryoprotection. Tissue was embedded in OCT compound (Tissue-Tek), frozen and sectioned on the cryostat (Leica) at 12 µm (DRG) or 20 µm (spinal cord). Tissue sections were permeabilized and blocked in 8% bovine serum albumin (BSA), 1% Triton X-100 in PBS for 1 h at room temperature (RT), followed by overnight incubation at room temperature with primary antibodies diluted in 2% BSA, 0.3% Triton X-100 and PBS. Secondary antibodies (Invitrogen) were incubated for 2 h at RT. All sections were counterstained with Hoechst (Molecular Probes). To quantify and assess the fluorescent intensity, all images for individual experiments were taken using the same optical parameters. Fluorescence measurements were performed using FIJI software, in GFP+ cells, normalized against the background.

## Quantification of axonal growth

Spinal cord micrographs were taken at 10x magnification for analysis on the Leica SP8 confocal microscope (Leica). To quantify axons, images were scaled, and the grid overlay plugin was used within FIJI to measure fluorescence intensity at 100 µm increments from the lesion border site and normalized to signal intensity caudal to the lesion site. An inter-animal comparable ratio was obtained by normalizing the number of labeled axons at each distance to the total number of labeled axons at 1 mm caudal to the lesion border. To measure the longest axon, the distance from the lesion border site to the longest axon was measured. To measure caudal axon retraction, the distance from the lesion border site to the main retraction bundle was measured. All analysis was done using FIJI.

## DRG culture and viral overexpression

Twenty-four-well plates were treated with 0.1 mg/ml PDL and 2 µg/ml laminin. DRG were dissected and collected in Hank's buffered saline solution (HBSS) on ice. HBSS was aspirated, and DRG enzymatically digested for 45 min at 37 °C with Dispase II (5.0 mg/ml) and Collagenase Type II (2.5 mg/ml) in Dulbecco's modified Eagle's medium (DMEM). Digest media was aspirated and replaced with 10% heat-inactivated fetal bovine serum (FBS), 1x B27 in DMEM:F12 to wash cells. DRG were then dissociated by pipetting to disaggregate tissue and release cells. Cells were then spun down and resuspended in 1% penicillin/streptomycin and 1x B27 in DMEM:F12. Cells were plated at a density of 4000 cells/well and maintained at 37 °C with 95% $O_2$ and 5% $CO_2$. AAV-GFP or AAV-Cited2-GFP (at a concentration of $10^{13 \text{ vg/ml}}$) was added to each well and incubated for 5–7 days.

## Embryonic DRG culture and viral overexpression

Twenty-four-well plates were treated with 20 µg/ml PLL and 5 µg/ml laminin. Embryonic DRG culture was carried out as previously described (Nascimento et al, 2022). DRG were extracted from E13 mouse embryos, then placed into 0.05% trypsin-EDTA digest solution for 1 h at 37 °C. Neurobasal with 15% FBS was added to stop digest, then media aspirated, and DRG resuspended in 2% B27, 2 mM L-glutamine, 1% penicillin-streptomycin, 50 ng/ml of nerve growth factor in Neurobasal media. DRG were dissociated by pipetting to disaggregate tissue and release cells. Cells were plated at a density of 60,000 cells/well and maintained at 37 °C with 95% $O_2$ and 5% $CO_2$. Glial-absent monocultures were obtained by exposing cells to 40 µM uridine and 40 µM 5-fluoro-2'-deoxyuridine 1 day post-plating. For viral overexpression, AAV-GFP or AAV-Cited2-GFP (at a concentration of $10^{10 \text{ vg/ml}}$) was added to each well at DIV3. Cells were fixed in 4% PFA at DIV20 for immunocytochemistry.

## Neurite outgrowth assays

DRG cells were extracted as described above. Following dissociation, cells were spun down, washed and resuspended in electroporation buffer (Buffer R). Cells were electroporated using the Neon Transfection System (Thermo Fisher Scientific), then resuspended in 1% penicillin/streptomycin and 1x B27 in DMEM:F12 and plated at a density of 25,000 cells/well. For loss-of-function experiments, cells were electroporated with GFP and CRISPR-Cas9 Cited2 double nickase or GFP and CRISPR-Cas9 double nickase control plasmids and then cultured on well plates treated with 0.1 mg/ml PDL and 2 µg/ml laminin for 24 h. For gain-of-function experiments, cells were electroporated with control pcDNA3.1GFP or pcDNA3.1Cited2 overexpression plasmid tagged with GFP (human Cited2 gene with GFP tag fused at the amino terminus modified from #48184, Addgene) and then cultured on well plates treated with 0.1 mg/ml PDL and 2 µg/ml laminin for 24 h. Alternatively, they were cultured on plates coated with 0.1 mg/ml PDL and 2 µg/ml laminin or 4 µg/cm² rat myelin and incubated for 24 h.

## F11 cell culture

F11 cells, a somatic cell hybrid of rat embryonic RGS and mouse neuroblastoma cell line N18TG2, were cultured until 95% confluent in 1% penicillin/streptomycin and 1x B27 in DMEM:F12 at 37 °C with 95% $O_2$ and 5% $CO_2$. Media was aspirated, cells spun down, washed, and resuspended in electroporation buffer. Cells were electroporated with control GFP (pcDNA3.1GFP) or pcDNA3.1-Cited2 overexpression plasmid tagged with GFP, as described above, and plated at a density of 25,000 cells/well. Wells were coated with 0.1 mg/ml PDL and 2 µg/ml laminin and incubated for 72 h. Cells were then prepared for immunoblotting.

## Immunocytochemistry

Adult DRG cultured cells were fixed in 4% PFA, washed with PBS, then permeabilized and blocked in 2% BSA, 0.3% Triton-X in PBS for 1 h at RT. Cells were incubated with primary antibodies in 2% BSA, 0.1% Triton-X in PBS overnight at RT. Cells were then incubated with secondary antibodies for 1 h at RT, and Hoechst and coverslips were mounted with Mowiol mounting medium.

Embryonic DRG cultured cells were fixed in 4% PFA and washed in PBS. After fixation, cells were immediately permeabilized

and blocked with 0.3% Triton X-100 and 2% BSA in PBS for 1 h at RT. Cells were incubated with primary antibodies diluted in blocking solution overnight at RT, then incubated with secondary antibodies diluted in blocking solution for 1 h at RT and Hoechst.

## DRG culture image quantification

For quantification of neurite outgrowth, total neurite outgrowth was quantified by measuring all cells with outgrowth and averaging across the well. All cells with outgrowth greater than their soma were assessed, and the average neurite outgrowth/cell was quantified per well/condition. Quantification was performed using Neuron J software. ROIs were selected from GFP+ cells only. To quantify and assess the fluorescent intensity, all images for individual experiments were taken using the same optical parameters. Thresholding was applied to subtract background immunofluorescence from analysis. Fluorescence measurements were performed using FIJI software, in GFP+ cells, normalized against the background. To assess embryonic DRG morphology, ~100 random cells were analysed per condition. Cells were visualized under a fluorescence microscope (Leica DMI6000 B) with a 40x / 0.60 NA objective coupled to a camera (Hamamatsu C11440-22c). The analysis was done on cells where either AAV-Cited2-GFP or AAV-GFP control was overexpressed. They were classified as being multipolar (with more than two neurites), bipolar (two neurites with a spindle shape), bell-shaped bipolar (two neurites approach each other forming an angle of less than 90 degrees) and pseudo-unipolar (a single stem neurite that bifurcates into two neurites).

## Real-time quantitative PCR (RT-qPCR) and reverse transcription PCR

Total RNA extraction from sciatic DRG was done using the RNeasy Mini Kit. DRG were dissected as described earlier and placed into Buffer RLT with beta-mercaptoethanol. Tissue was then washed with 70% ethanol, followed by Buffer RW1 and RPE. Finally, samples were eluted in 30 μl RNase-free water. RNA concentration and purity was measured using a nanodrop. RNA was treated with DNase I, followed by cDNA synthesis using SuperScript First-Strand Synthesis System kit: 1 μg of RNA was incubated with Oligo $(dT)_{12-18}$ (0.5 μg/μl), random hexamers (50 ng/μl), 10 mM dNTP mix for 5 min at 65 °C. Then, 10x cDNA synthesis buffer, 25 mM $MgCl_2$, 0.1 M DTT, and RNaseOUT (400 U/μl) were added and incubated at RT for 2 min. Samples were incubated with Superscript II Reverse Transcriptase (50 U/μl) at RT for 10 min, followed by 42 °C for 50 min. Reaction was terminated by incubating samples at 70 °C for 15 min. Samples were placed on ice and incubated with *E.coli* RNase H (2U/μl) at 37 °C to remove RNA complementary to cDNA. RT-qPCR was performed using KAPA HiFi HotStart ReadyMix. 3 μl of template was added to 0.5 μM of forward and reverse primer on the Applied Biosystems 7900HT FAST (Thermo Fisher Scientific) detector. Thermal-cycling conditions were as follows: 95 °C for 5 min; followed by 45 cycles of 95 °C for 20 s, 56 °C for 20 s, 72 °C for 20 s; and melting curve analysis. Data acquisition occurred at 72 °C and at 1 °C increments during melting curve analysis. For analysis of data, $2^{-\Delta CT}$ method was used. PCR was performed by using 15 ng of cDNA and run with the following conditions: 95 °C for 5 min, followed by 30 cycles of 94 °C

for 30 s, 54 °C for 30 s, and 72 °C for 1 min; and extension at 72 °C for 10 min, and then analysed by agarose gel electrophoresis using SYBR green to visualize bands.

## Immunoblotting

Protein extracts from sciatic DRG tissue, DRG cell culture, or F11 in vitro cell culture were extracted using RIPA buffer supplemented with protease and phosphatase inhibitor cocktails (Roche). Lysates were sonicated using a tip sonicator for 2 × 30 s on/off and were incubated for 30 min followed by centrifugation at 4 °C. The concentration of protein lysates was determined using the Pierce BCA Protein Assay Kit. Protein was loaded onto 10–15% SDS-PAGE gels. Membranes were blocked with 5% milk for 1 h RT and incubated with primary antibodies at 4 °C ON. Membranes were incubated with HRP-linked secondary antibodies (GE Healthcare) for 1 h at room temperature and developed with ECL plus substrate (Thermo Fisher Scientific). Band densitometry was quantified using ImageJ software.

## Neuronal enrichment

One animal was used per condition (AAV-GFP control or AAV-Cited2-GFP). Twenty-four hours after injury, L4-6 DRG were dissected and collected in HBSS on ice. HBSS was aspirated, and DRG digested for 30 min at 37 °C with Dispase II (1.0 mg/ml), Collagenase Type II (2.5 mg/ml), 1:150 Papain in DMEM with GlutaMAX. Digest solution was aspirated, and cells were washed in 10% heat-inactivated FBS, 1x B27, 1:20 SUPERase In RNase Inhibitor 20 U/μl (Thermo Fisher Scientific) in DMEM/F12, with GlutaMax. Cells were then mechanically dissociated by gentle pipetting, first using a glass Pasteur pipette, then using a P1000 pipette. Cells were strained through a 70-μm strainer to obtain single cells and remove debris/clumps. The total collection volume following dissociation was ~1.5–2 ml per mouse. To prepare the gradient; 15 ml Falcons containing 15% fresh BSA in DMEM were placed on ice. Cells were then gently added on top of gradient, being careful to avoid mixing the solutions together. Cells and gradient were then centrifuged at 80×g for 8 min at 4 °C. Supernatant was aspirated carefully and the cell pellet was resuspended in an appropriate solution. To confirm neuronal enrichment, cell samples before and after the gradient were placed into culture and imaged 2 h or 3 days post-plating or prepared for PCR.

## RNA-sequencing

Following the neuronal gradient, the cell pellet was resuspended in buffer RLT with beta-mercaptoethanol and transferred to a 1.5 ml DNA LoBind Eppendorf (Eppendorf). RNA was extracted using the RNeasy Micro kit following the protocol supplied with minor modifications. Lysate was homogenized by vortexing for 1 min, followed by the addition of 70% ethanol. The sample was centrifuged, flow-through discarded and Buffer RW1 added. The sample was centrifuged again, followed by on-column DNase I treatment. Samples were incubated with Dnase I in Buffer RDD for 15 min at RT, followed by the addition of Buffer RW1. Samples were centrifuged, then 80% fresh ethanol was added, and the samples were centrifuged again. Finally, samples

were eluted into a 1.5 ml DNA LoBind Eppendorf with RNase-free water. RNA quality and concentration was determined using an RNA 6000 Pico Assay on the 2100 Bioanalyzer (Agilent). RNA with a quality RIN factor above 8 was used for sequencing.

Libraries were prepared using the TruSeq Stranded mRNA preparation kit using the low sample protocol. Poly(T)-attached magnetic beads were used for poly(A) RNA enrichment. This was followed by reverse transcription and library preparation. Sequencing was performed using Illumina HiSeq 4000 75-base-pair (bp) paired-end sequencing to obtain ~60 mil reads per sample. Sequences were demultiplexed, and adapters trimmed with bcl2fastq-v2.20 (Illumina) and read quality control was carried out using FastQC-v0.11.9 and trimming of reads less than Phred33 quality score 20 or remaining adapters with TrimGalore!-v0.6.6. RNA-seq analysis was run using the COMBINE lab's Salmon-DESeq2 pipeline. Salmon-v1.6.0 was used in mapping-based mode to quantify read sets. Approximately 92% of reads were mapped to the M25 GENCODE reference mouse genome. Salmon's output files were imported and converted for DESeq2 using Tximeta-v1.12.4, followed by differential expression analysis using DESeq2-v1.34.0 with R-v4.1.2. Normalized counts were generated using the median of ratios method in DESeq2. The dataset is available at NCBI GEO under the accession number GSE214722.

## ATAC-sequencing

ATAC-seq was performed as previously reported (Müller et al, 2023). Following the neuronal gradient, the cell pellet was resuspended in lysis buffer. Transposition reaction (Illumina) was carried out 10,000. DNA was purified using the PCR MinElute kit (QIAGEN) following the manufacturer's instructions. After elution, DNA was amplified by 12-cycle PCR using barcoded primers. Samples were then purified using AMPure XP Beads (Beckman Coulter) following the manufacturer's instructions using a left side selection (1.6x ratio). Libraries were run on the 2100 Bioanalyzer to determine DNA quality and size using a high-sensitivity DNA Assay and multiplexed. Sequencing was run using HiSeq 4000 (Illumina) 75 bp paired-end sequencing to obtain ~80 million read pairs per sample. Sequences were demultiplexed and adapters trimmed with bcl2fastq. Quality control, read alignment, signal track generation, and peak calling was carried out using the Kundaje lab's ATAC-seq processing pipeline running Bowtie2 and MACS2 (https://github.com/kundajelab/atac_dnase_pipelines). Methods described previously (Palmisano et al, 2019) were used to examine differentially accessible genes at the promoter and enhancer regions. For promoter analysis, genomic bins 1000 bp upstream and downstream of the transcription start site (TSS) were created. Gene counts per genomic bin were obtained from the mapped reads using HTSeq-v0.6.1. For enhancer analysis, read counts per peak were obtained from the mapped reads using HTSeq-v0.6.1. Then, differential accessibility analysis was carried out using EdgeR-v3.8.6. To determine the enrichment compared to the background, the GenomicAlignment-v1.32.0 package was used to calculate the number of reads within each sample divided by the proportion of the genomes spanned by that type of region. Signal distribution plots were created using NGSplot-v2.47.1.

## Footprint and transcription factor analysis

ATAC-seq footprinting was performed using the TOBIAS suite of tools (Bentsen et al, 2020). To identify footprints within the differentially accessible regions (both promoter and enhancer), replicates for the Cited2 and GFP conditions were first merged. Single-base-resolution cut-site tracks were generated with TOBIAS ATACorrect, and continuous footprint scores were calculated across genomic regions using TOBIAS ScoreBigWig. Footprints were then associated with differentially accessible promoters (±1 kb) or enhancers (±0.5 kb).

Transcription factor (TF) activity dynamics were compared between Cited2 or GFP conditions using TOBIAS BINDetect, with motif matching performed against the JASPAR 2020 vertebrate non-redundant database. To further examine TF activity, Bivariate Genomic Footprinting (BaGfoot analysis was performed for promoters or enhancers.

## CUT&TAG-qPCR

Panobinostat (20 mg/kg; Selleckchem, UK) or vehicle control (DMSO) was administered systemically via intraperitoneal injection. After 3 h, DRGs were dissected, and total DRGs from two mice were pooled together per biological replicate. Samples were immediately flash-frozen in liquid nitrogen and stored at −80 °C.

All tubes (Eppendorf and Falcon) were pre-coated with 1% BSA 1 day prior to nuclei isolation to minimize wall adherence. Frozen DRGs were transferred from −80 °C into homogenization buffer (0.25 M sucrose, 25 mM KCl, 5 mM MgCl₂, 20 mM Tricine-KOH, pH 7.8, 5 µg/mL actinomycin, 1% BSA, 0.15 mM spermine, 0.5 mM spermidine, EDTA-free protease inhibitor, phosphatase inhibitors, and 0.1% Triton X-100). After brief incubation on ice, tissue was homogenized using a pellet pestle (Kimble) for 15 s, followed by rinsing the pestle with homogenization buffer. Homogenates were allowed to settle, then filtered sequentially through 70 µm and 40 µm filters, repeating as necessary.

Nuclei were then incubated with anti-NeuN Alexa Fluor™ 488 antibody (Merk, MAB377X; 1:100) for 2 h at 4 °C. Samples were washed twice in homogenization buffer (1100×$g$, 5 min). Pellets were resuspended in 1.5 mL homogenization buffer containing DAPI (1:1000; 0.1 mg/mL). DAPI⁺/NeuN⁺ nuclei were sorted on a FACSAria II cell sorter directly into 100 µL homogenization buffer. A total of 100,000 nuclei were collected per sample.

Bench-top CUT&Tag was performed as described previously (Kaya-Okur and Henikoff, 2019) with minor modifications. Sorted nuclei were mildly fixed in 0.5% paraformaldehyde for 1 min at room temperature and quenched with glycine (125 mM final concentration) for 5 min. Nuclei were washed twice in PBS and resuspended in 100 µL wash buffer (20 mM HEPES, pH 7.6, 150 mM NaCl, 2 mM EDTA, and 0.5 mM spermidine). Concanavalin A-coated magnetic beads (Bangs Laboratories, BP531) were prepared as previously described, and 10 µL activated beads were added per sample in PCR strip tubes, followed by incubation at room temperature for 10 min. Beads were placed on a magnetic rack, supernatant was removed, and bead-bound nuclei were resuspended in 100 µL antibody buffer (20 mM HEPES, pH 7.6, 150 mM NaCl, 2 mM EDTA, 0.5 mM spermidine, 0.05% digitonin, protease inhibitor, and 1% BSA) for 1 h at room temperature to block. Primary antibody against H3K27ac (Abcam ab177178; 1:50)

was added and incubated overnight at 4 °C with shaking on a thermal mixer (600 rpm). After magnetic clearing and removal of unbound primary antibody, guinea pig anti-rabbit IgG secondary antibody (Antibodies online, ABIN101961, 1:100 in Dig-wash buffer) was added for 1 h at room temperature. Nuclei were washed three times with Dig-wash buffer. Samples were resuspended in 50 µL Dig-300 buffer (20 mM HEPES pH 7.5, 300 mM NaCl, 0.5 mM spermidine, protease inhibitor, 0.05% digitonin), and CUTANA™ pAG-Tn5 (1:150; Epicypher 15-1017) was added. Binding proceeded for 1 h at room temperature with shaking (800 rpm). After three Dig-wash buffer washes, nuclei were resuspended in 50 µL tagmentation buffer (10 mM $MgCl_2$ in Dig-300 buffer) and incubated at 37 °C for 1 h.

To terminate tagmentation, 100 µL Proteinase K/SDS solution (0.5 mg/mL Proteinase K, 0.5% SDS, 10 mM Tris-HCl, pH 8.0) was added. Samples were vortexed briefly (2 s) and incubated at 55 °C for 1 h. After centrifugation, supernatants were cleared on a magnetic rack, transferred to new tubes, and processed using the Zymo DNA Clean & Concentrator kit (Zymo Research D4013), eluting in 23 µL elution buffer.

For PCR amplification, 21 µL purified DNA was combined with 2 µL universal i5 and uniquely barcoded i7 primers, and 25 µL NEBNext HiFi 2× PCR Master Mix. Samples were cycled as follows: 72 °C 5 min; 98 °C 30 s; 18 cycles of 98 °C 10 s, 63 °C 30 s; 72 °C 1 min; hold at 8 °C. Libraries were cleaned using 1.3× Ampure XP beads (Beckman Coulter) with two 80% ethanol washes and eluted in 25 µL 10 mM Tris-HCl, pH 8.0.

H3K27ac CUT&TAG library was used to quantify the relative fold signal at the H3K27ac sites using Cited2 promoter-specific primers (TSS1, TSS2, Middle, Distal) by using SYBR-based qPCR. Reactions contained 1 µL library, 10 µL KAPA SYBR FAST qPCR Master Mix (KK4601), 0.4 µL each of 10 µM forward and reverse primers, 0.4 µL ROX Low, and nuclease-free water to 20 µL. The cycling program was: 95 °C 5 min; 39 cycles of 95 °C 20 s, 60 °C 30 s, 72 °C 45 s; followed by a melt curve up to 90 °C. Relative enrichment was calculated using the ΔΔCt method, comparing four Cited2 promoter H3K27ac-positive regions to the GAPDH H3K27ac-positive promoter as an internal control.

## Gene Ontology (GO) and KEGG analysis

Gene Ontology (GO) and KEGG pathway analysis was carried out using DAVID (Database for Annotation, Visualization, and Integrated Discovery)-v2021 (Huang et al, 2009) and ClueGo via Cytoscape-v3.9.1 for all differentially expressed or accessible genes using all expressed or accessible genes within the DRG as background. Differentially expressed or accessible genes were only considered if $p < 0.05$ and >4 genes were enriched for each term. GO and KEGG terms were filtered by $p < 0.01$. Graphs and dot plots were made using Graphpad Prism v9.4.0 or using ggplot2 v3.3.6 in R, respectively.

## Dataset correlation analysis

Raw fastq files from previously published datasets (Tedeschi et al, 2016; Data Ref: Tedeschi et al, 2016; Palmisano et al, 2019; Data Ref: Hervera et al, 2018; Data Ref: Southard-Smith et al, 2014) were extracted from NCBI by using the SRAtoolkit-v2.11.2 and submitted to the same Salmon-DESeq2 pipeline to permit valid

### The paper explained

**Problem**

Spinal cord injury is a neurological condition characterized by permanent impairment in neurological function, causing long-term disability. Current treatments rely mainly on neurorehabilitation and neurostimulation, that at best lead to partial clinical improvement in incomplete lesions. There is an urgent need for alternative strategies that restore the regenerative potential of the injured spinal cord.

**Results**

We show that manipulation of Cited2 can reprogram sensory neurons towards an immature state and enable regeneration in a mouse model of spinal cord injury. The use of Cited2 targeting HDAC inhibitor Panobinostat promotes sensory and motor neuron growth and sprouting as well as functional recovery after spinal cord injury.

**Impact**

Our findings identify a novel possible therapeutic use for clinically established Panobinostat in spinal cord injury. By inhibiting histone deacetylases, repurposing this drug offers a realistic route toward clinical translation, with the potential to improve outcomes for patients with spinal cord injuries.

comparisons between datasets. Correlation analysis (Fisher's exact test, odds ratio) was carried out using the GeneOverlap package and Pearson's correlation analysis in R for differentially expressed genes with $p < 0.05$ and/or FDR <0.05. Integrated ATAC-seq and RNA-seq correlation analysis was carried out following previously published methodology (Starks et al, 2019) to examine the correlation between gene expression and promoter accessibility of genes after Cited2 overexpression. Venn diagrams showing gene overlaps were created using BioVenn and Adobe Illustrator.

## Quantification and statistical analysis

All experiments were performed in a blind manner following systematic randomization of the samples. The appropriate N for each experiment was selected following power calculations. Two-tailed Student's $t$-tests were performed for comparing a single variable between two groups (or Welch's $t$-test in the case of a non-normal distribution), one- or two-way analysis of variance (ANOVA) with Sidak/Tukey's multiple-comparison post hoc test, or a mixed-effects ANOVA with uncorrected Fisher's LSD post hoc test. All values are reported as mean ± standard deviation (SD). Sample sizes are indicated in figure legends, and significance was defined as *$p < 0.05$, **$p < 0.01$, ***$p < 0.001$, and ****$p < 0.0001$. ns not significant.

## Graphics

Graphics were created with BioRender.com.

## Data availability

All data were available in the main text or the Appendix. Cited2 overexpression versus GFP control RNA-seq and ATAC-seq datasets are deposited at GEO and are publicly available as of the date of publication under the accession numbers GSE214722 and

GSE224110, respectively. All materials or data requests should be addressed to the corresponding author. This paper analyses existing, publicly available data. The accession numbers for the datasets are listed in the methods. This paper does not report original code.

The source data of this paper are collected in the following database record: biostudies:S-SCDT-10_1038-S44321-026-00385-w.

## Peer review information

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

## Acknowledgements

This work was supported by Wings for Life (SDG); The Rosetrees Trust (SDG); Spinal Research (SDG); Brain Research Trust (FM), Dr. Miriam and

Sheldon G Adelson Medical Research Foundation (SDG); the National Institute for Health Research (NIHR) Imperial Biomedical Research Centre (MED, SDG); by the National Institute Of Neurological Disorders And Stroke of the National Institutes of Health (IP, Award Number R56NS138373); Wings for Life (WFL-US-20/24 IP) The views expressed are those of the author(s) and not necessarily those of the NHS, the NIHR, the NIH, or the Department of Health. The Imperial BRC Genomics Facility has provided resources and support that have contributed to the research results reported within this paper. The Imperial BRC Genomics Facility is supported by NIHR funding to the Imperial Biomedical Research Centre. We would like to thank Dr. Matt Danzi for providing guidance on computational analysis.

## Author contributions

**Franziska Müller**: Conceptualization; Formal analysis; Investigation; Methodology; Writing-review and editing. **Eilidh Maclachlan**: Formal analysis; Investigation; Methodology. **Ana Catarina Costa**: Formal analysis; Investigation; Methodology. **Jia Qu**: Formal analysis; Methodology. **Bishal Shrestha**: Formal analysis; Methodology. **Zheng Wang**: Formal analysis. **Francesco De Virgiliis**: Investigation; Methodology. **Thomas Haynes Hutson**: Formal analysis; Investigation; Methodology. **Luming Zhou**: Investigation; Methodology. **Guiping Kong**: Investigation; Methodology. **Jessica S Chadwick**: Formal analysis; Investigation. **Paolo La Montanara**: Investigation. **Zhulin Yuan**: Investigation. **Nejc Haberman**: Formal analysis. **Monica M Sousa**: Data curation; Methodology; Project administration. **Ilaria Palmisano**: Conceptualization; Investigation; Formal analysis; Methodology; Review and editing; Funding. **Simone Di Giovanni**: Conceptualization; Data curation; Supervision; Funding acquisition; Writing—original draft; Project administration; Writing—review and editing.

Source data underlying figure panels in this paper may have individual authorship assigned. Where available, figure panel/source data authorship is listed in the following database record: biostudies:S-SCDT-10_1038-S44321-026-00385-w.

## Disclosure and competing interests statement

The authors declare no competing interests.

# Expanded View Figures

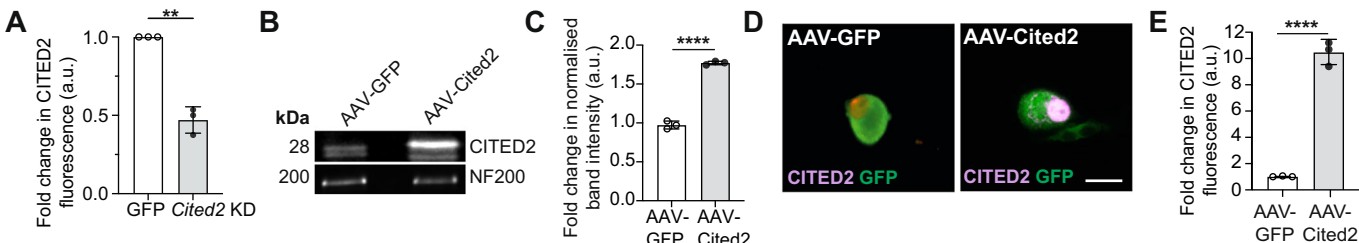

**Figure EV1.  Cited2 downregulation and overexpression.**

(A) Quantification of Cited2 knockdown in cultured DRG neurons (*t*-test, *p* = 0.0092, *n* = 3 independent biological replicates, about 50 cells per replicate). (B, C) Confirmation and quantification of CITED2 overexpression efficiency following AAV-GFP or AAV-Cited2-GFP in DRG cultured neurons using immunoblotting analysis. Band intensity normalized against NF200 (*t*-test, *p* = 1.492 × 10$^{-5}$, *n* = 3 independent biological replicates). (D, E) Confirmation and quantification of CITED2 overexpression (magenta) in GFP$^{+}$ DRG cultured neurons after addition of AAV-GFP or AAV-Cited2-GFP (*t*-test, *p* = 4.516 × 10$^{-5}$, *n* = 3 independent biological replicates, about 50 cells per replicate). Scale bar: 20 µm. *\*p* < 0.05, *\*\*p* < 0.01, *\*\*\*p* < 0.001, and *\*\*\*\*p* < 0.0001. ns not significant. All error bars shown as standard deviation (SD).

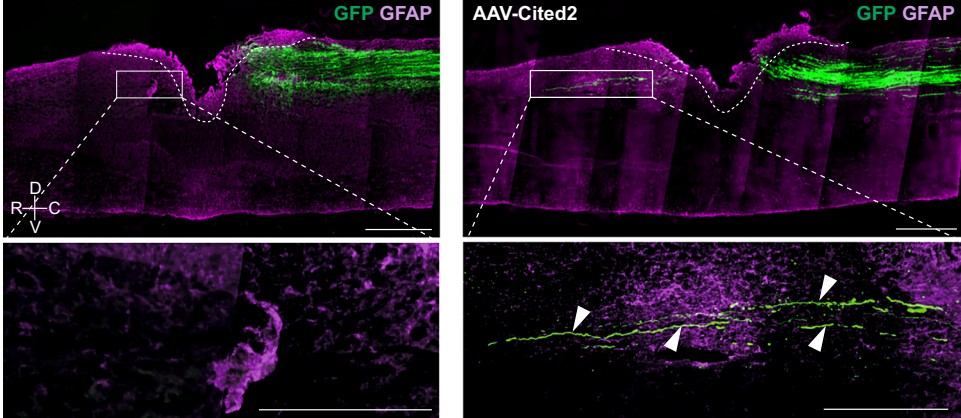

**Figure EV2.    Additional representative micrographs of AAV-GFP or AAV-Cited2-GFP overexpression.**

AAV-Cited2-GFP promoted axon growth into and past the lesion site (green, white arrows) 6 weeks post-SCI. GFAP (magenta) was used to determine the lesion site (white dotted line). Scale bar: 200 µm. Scale bar zoomed inset: 100 µm. Source data are available online for this figure.

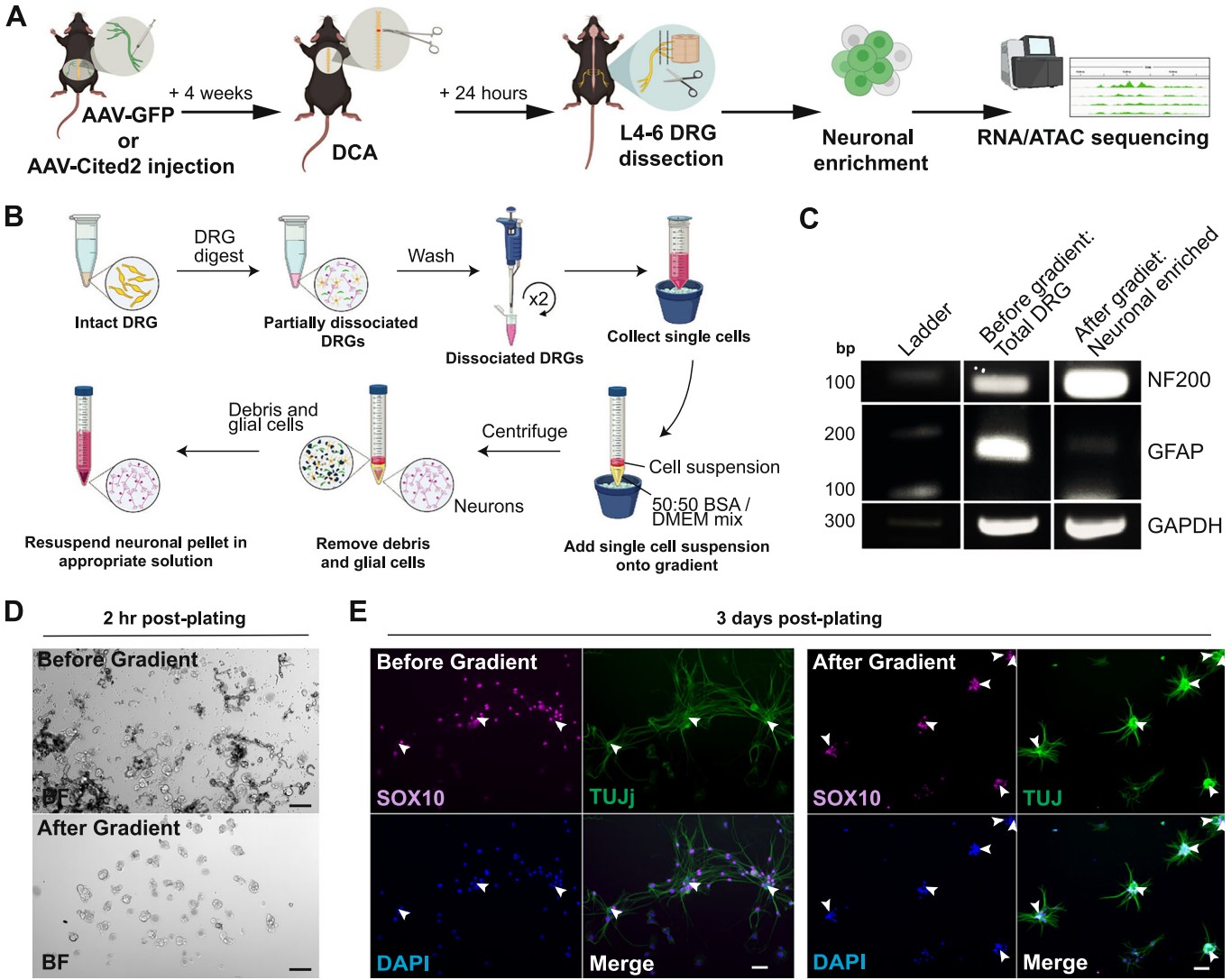

**Figure EV3. Confirmation of neuronal enrichment following gradient separation.**

(A, B) Experimental design (made with BioRender). (C) mRNA expression as determined by reverse transcription (RT)-PCR with neuronal markers (NF200) or glial markers (GFAP) before and after neuronal gradient separation. GAPDH was used to normalize band intensity. (D) Representative brightfield micrographs of DRG cells before and after neuronal gradient. (E) SOX10 (magenta) and beta 3 tubulin (TUJ, green) immunostaining in DRG cells before and after neuronal gradient. Scale bar: 50 μm.

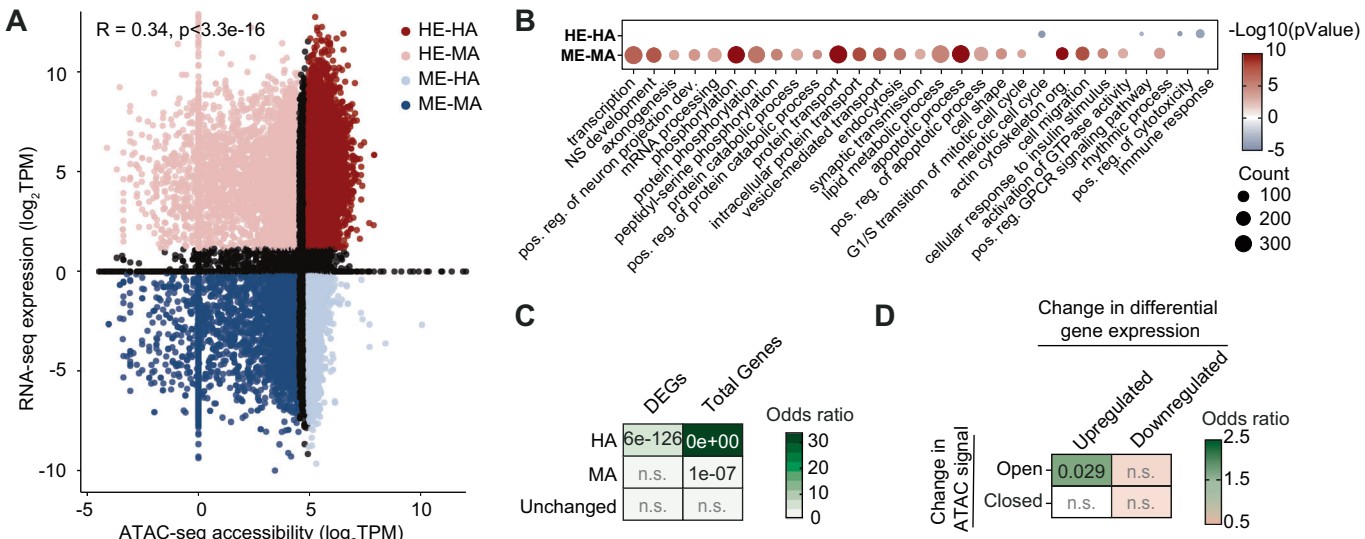

**Figure EV4. Gene expression and chromatin accessibility after Cited2 overexpression and SCI.**

(A) Correlation analysis between gene expression (RNA-seq) and chromatin accessibility (ATAC-seq) following Cited2 overexpression [Spearman correlation $p$ value $< 3.714 \times 10^{-7}$, $R^2 = 0.34$] using log2 of normalized TPM. Genes are split into four categories: highly expressed and highly accessible (HE-HA 7582 genes) if their normalized TPM is higher than the 75th percentile of the data; low-to-medium expression and low-to-medium accessibility (ME-MA 6420 genes) if their TPM is lower than the 50th percentile of the data; highly expressed and low-to-medium accessibility (HE-MA 4336 genes); and low-to-medium expression and high accessibility (ME-HA 2396 genes). Black dots indicate genes that do not fall into any category. (B) GO and KEGG analysis of HA-HE or ME-MA genes. $P$ values are calculated using a modified Fisher's exact test as described in DAVID (Huang et al, 2009). (C) Odds ratio and Fisher's exact tests between differentially expressed (DEGs, $p < 0.05$) or non-differentially expressed (Total genes, $p > 0.05$) genes and genes showing high accessibility (HA), low-to-medium accessibility (MA), or unchanged accessibility (UA). (D) Odds ratio and Fisher's exact tests between changes in differential gene expression (DEGs, $p < 0.05$) and changes in chromatin accessibility (ATAC signal, $p < 0.05$). Color represents correlation (odds ratio) while numbers represent significance (Fisher's exact test).

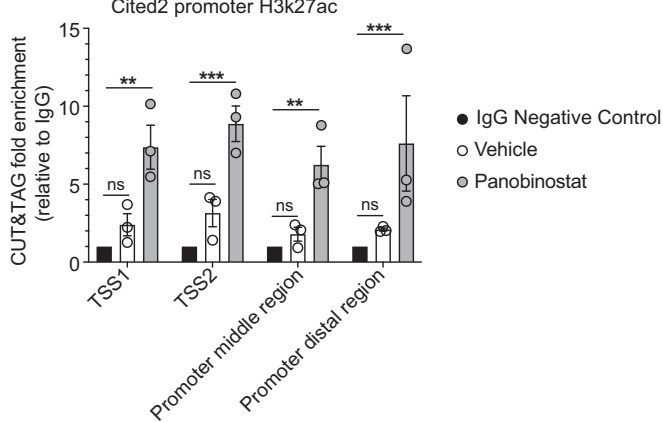

**Figure EV5. Panobinostat increases H3k27ac occupancy at the *Cited2* promoter in adult DRG neurons.**

CUT&Tag qPCR quantification of H3k27ac at two TSS sites (TSS1 and TSS2), the middle promoter region, and the distal promoter region. Data were expressed relative to the IgG-negative control. Two-way ANOVA with Dunnett's post hoc test (Group: $f(2,24) = 36$, $p = 5.010 \times 10^{-8}$; Region: $f(3,24) = 0.6$, $p = 0.5661$; Interaction score: $f(6,24) = 0.2$, $p = 0.9601$. Displayed is the Treatment $p$ value. IgG vs Panobinostat—TSS1, $p = -0.0011$; TSS2, $p = 0.0001$; middle promoter region, $p = 0.0063$; distal promoter region, $p = 0.0008$; IgG vs vehicle, not significant for all regions. $n = 3$ independent biological replicates per group. *$p < 0.05$, **$p < 0.01$, ***$p < 0.001$, and ****$p < 0.0001$. ns not significant. All error bars shown as standard deviation (SD).

