## [Peer Review File · EMBO Molecular Medicine]

Druggable CITED2 Orchestrates an Epigenetic Switch from Neuronal Maturation to Regenerative Decline

Simone Di Giovanni, Franziska Mueller, Eilidh Maclachlan, Ana Costa, Jia Qu, Bishal Shrestha, Zheng Wang, Francesco De Virgiliis, Thomas Hutson, Luming Zhou, Guiping Kong, jessica Chadwick, Monica Sousa, Ilaria Palmisano, Zhulin Yuan, Nejc Haberman, and Paolo La Montanara

Corresponding author: Simone Di Giovanni (s.di-giovanni@imperial.ac.uk)

Review Timeline:

Submission Date:	24th Jul 25
Editorial Decision:	12th Aug 25
Revision Received:	15th Dec 25
Editorial Decision:	9th Jan 26
Revision Received:	19th Jan 26
Accepted:	29th Jan 26

Editor: Jingyi Hou

Transaction Report:

13th Aug 2025

Dear Prof. Di Giovanni,

Thank you again for submitting your work to EMBO Molecular Medicine. We have now received the reports from the three reviewers and as you will see below, the reviewers think that the study is potentially interesting. They raise however a series of concerns, which we would ask you to convincingly address in a revision.

I think the referees' recommendations are clear and need not be repeated here. All issues raised by the referees need to be satisfactorily addressed. As you may already know, our editorial policy allows in principle a single round of major revision so it is essential to provide responses to the referees' comments that are as complete as possible. Please feel free to contact me in case you would like to discuss in further detail any of the issues raised by the referees.

Please also contact us as soon as possible if similar work is published elsewhere. If other work is published, we may not be able to extend the revision period beyond three months.

I look forward to receiving your revised manuscript soon.

Kind regards,
Jingyi

Jingyi Hou
Senior Editor
EMBO Molecular Medicine

We require:

- 1) A .docx formatted version of the manuscript text (including legends for main figures, EV figures and tables). Please make sure that the changes are highlighted to be clearly visible.
- 2) Individual production quality figure files as .eps, .tif, .jpg (one file per figure). For guidance, download the 'Figure Guide PDF': (<https://www.embopress.org/page/journal/17574684/authorguide#figureformat>).
- 3) A .docx formatted letter INCLUDING the reviewers' reports and your detailed point-by-point responses to their comments. As part of the EMBO Press transparent editorial process, the point-by-point response is part of the Review Process File (RPF), which will be published alongside your paper.
- 4) A complete author checklist, which you can download from our author guidelines (<https://www.embopress.org/page/journal/17574684/authorguide#submissionofrevisions>). Please insert information in the checklist that is also reflected in the manuscript. The completed author checklist will also be part of the RPF.
- 5) Please note that all corresponding authors are required to supply an ORCID ID for their name upon submission of a revised manuscript.

6) It is mandatory to include a 'Data Availability' section after the Materials and Methods. Before submitting your revision, primary datasets produced in this study need to be deposited in an appropriate public database, and the accession numbers and database listed under 'Data Availability'. Please remember to provide a reviewer password if the datasets are not yet public (see <https://www.embopress.org/page/journal/17574684/authorguide#dataavailability>).

12) Author contributions: You will be asked to provide CRediT (Contributor Role Taxonomy) terms in the submission system. These replace a narrative author contribution section in the manuscript.

13) A Conflict of Interest statement should be provided in the main text.

14) Please provide a 'Synopsis' to further enhance discoverability. Synopses are displayed on the journal webpage and are freely accessible to all readers. They include a short stand first (maximum of 300 characters, including space) as well as 2-5 one-sentences bullet points that summarizes the paper. Please write the bullet points to summarize the key NEW findings. They should be designed to be complementary to the abstract - i.e. not repeat the same text. We encourage inclusion of key

acronyms and quantitative information (maximum of 30 words / bullet point). Please use the passive voice. Please attach these in a separate file or send them by email, we will incorporate them accordingly.

Please also suggest a visual abstract to illustrate your article as a PNG file 550 px wide x 300-600 px high.

15) All Materials and Methods need to be described in the main text using our 'Structured Methods' format. According to this format, the Methods section includes a Reagents and Tools Table (listing key reagents, experimental models, software and relevant equipment and including their sources and relevant identifiers) followed by a Methods and Protocols section describing the methods, ideally using a step-by-step protocol format. The aim is to facilitate adoption of the methodologies across labs.

Please download and fill our Reagents and Tools Table template (.docx), which you can find in our author guidelines: <https://www.embopress.org/page/journal/17574684/authorguide#structuredmethods>

***** Reviewer's comments *****

Referee #1 (Comments on Novelty/Model System for Author):

The mature nervous system is not able to regenerate. Therefore, patients have to endure irreversible and permanent disabilities. So far, there is no cure for such insults. In the current work, Authors used the mouse nervous system as a model to decipher mechanisms underlying such discrepancies. By combining the analysis of available RNA sequences data with their own ATAC-seq and RNA seq analysis they uncovered a key factor limiting axon regeneration. They tested the involvement of Cited-2 in vitro and in vivo, showing its key role during axon regeneration. Furthermore, they uncovered a drug targeting this molecular pathway. The administration of this drug in injured mice, improve axon growth (through plasticity or/and regeneration) but also functional recovery. This work could lead to nervous system repair treatment to Human patients.

Referee #1 (Remarks for Author):

During development, axons grow over long distances to reach their postsynaptic targets. This growth capability is limited in time as reaching the postnatal stage, there is a shift from growth ability to connectivity (synapse formation and stabilization of the circuit). Most of adult axons do not grow anymore. However, in case of injury to the mature nervous system, inducing axon growth is key to establish and repair circuits. Therefore, it is critical to understand the molecular processes underlying such changes.

In the current work, Müller and colleagues compared the molecular state (RNA expression, chromatin state, promotor/enhancer accessibility) at key stages of neuronal development with neuronal state sustaining regeneration and when regeneration is abolished. To this end they used the very elegant model of dorsal root ganglia neurons. These neurons are very unique as they don't have any dendrites but present 2 axons: one projecting to the spinal cord, forming the dorsal column and the other one to the periphery and forming for example the sciatic nerve. While the peripheral branch is able to regenerate after an injury, the central branch has no regrowth ability.

Using this model, Authors show that when regeneration is induced (upon sciatic nerve injury), neuronal programs revert to a developmental state when axons are growing. This molecular plasticity is not triggered with dorsal column lesion. Interestingly one key factor underlying this mechanism is Cited-2 that is a transcription co-factor. Its manipulation in vitro and in vivo induces modulated regeneration. Indeed, its overexpression could induce dorsal column regeneration.

Interestingly, Cited-2 has been found as a target for Panobinostat as drug that could be used in cases of nervous system injury. Its administration in vivo is able to improve axon growth through regeneration and/or sprouting. These results lead to better functional outcome.

This work is very thorough and experiments are well executed.

There are some minor comments/suggestions:

P4: "Importantly, this process is not irreversible": the double negation does not read well. Would it be possible to rephrase? Could Authors explain the choice of dorsal column hemi section for the Panobinostat experiments instead of dorsal column crush as for the rest of the work?

Referee #2 (Comments on Novelty/Model System for Author):

This study reveals that CITED2 reverses the neuronal maturation process through epigenetic reprogramming, restoring the axonal regenerative capacity of mature dorsal root ganglion (DRG) neurons. The article first identifies a developmental-regeneration correlation window. It was discovered that the embryonic stage E12.5 (unpolarized stage) shares 190 upregulated genes (e.g., related to transcriptional regulation and cell cycle) with regenerative injury (sciatic nerve axotomy, SNA), while the mature stage E17.5 (polarized stage) is enriched with downregulated synaptic-related genes. This suggests that the loss of regenerative capacity coincides with the polarization process.

CITED2 exhibits high expression at E12.5 and after SNA, but not after non-regenerative spinal cord dorsal column axotomy (DCA). Overexpression of CITED2 recruits EP300/GBP, increases H3K9ac/H3K27ac, opens chromatin (e.g., increased accessibility in 2615 enhancer regions), and activates developmental transcription factors such as bHLH and HOX (e.g., Meis2, HoxD13). This reprograms mature neurons to an E12.5-like state. Furthermore, the clinically approved HDAC inhibitor Panobinostat was found to promote axonal regeneration (its pro-growth effect disappears after Cited2 knockdown) and functional recovery (improved gridwalk test performance, increased 5-HT innervation) through a CITED2-dependent mechanism. The topic is very interesting and most experiments are properly done. However, I have some doubts on the presentation of the manuscript and on the conclusions reached by the authors based on these experiments. Additional data should be provided to sustain the discussion. Overall, the research showed plenty of phenotypes and data analysis related to neuronal maturation process and epigenetic reprogramming, however, the quality of the data in the article is not high and need improvements. And there are multiple format errors, especially in the data showing.

Major comments:

1. What is the logic from Fig.1 to Fig.2? that means what is logic for narrowing down to Cited2? It seems that there is no correlation between data analysis and Cited2?
2. The image quality of Fig. 2 needs improvement. Panels are low-resolution and some are missing clear panel labels. Replace with higher-resolution images, ensure all panels are labeled consistently, and match figure numbers in text and legends.
3. In Fig.9, how to distinguish the newly regenerated axons from degenerating fibers. Better to use tracer injections or genetic lineage labeling to test. Or quantify axons by counts/length rather than fluorescence intensity ratios.
4. The mechanism by which Panobinostat elevates CITED2 expression remains unelucidated. While the paper mentions that Panobinostat increases histone acetylation, it does not validate whether this directly affects the CITED2 promoter.
5. CITED2 regulates cell cycle genes (e.g., CDK1) and is aberrantly expressed in tumors. The paper does not assess the potential negative effects of Panobinostat on other cells, such as whether it could cause abnormal proliferation or tumorigenesis?
6. Behavioral assessment strategy (Fig 9E), BBB/BMS scores are standard for SCI locomotor assessment, why the authors only did Gridwalk test? Better to test with BBB/BMS scores.
7. Here the author showing data that CITED2-overexpression upregulated a batch of genes, more accessible chromatin, like confirmed Cbx5 and HoxD3, however no direct evidence to show how these genes show more accessible chromatin? Could CITED2 bind directly and promote accessible chromatin?

Minor comments:

There are many errors in the manuscript, including spelling, tense, figure legends, and data analysis methods, and it must be double checked very seriously throughout the manuscript to ensure that all errors are corrected.

1. The image quality of Fig. 2 needs improvement. The figure legend misses the label "Fig 2". In Fig. 8C, the text "GFP" in the top image appears obscured. In Fig. 8E, the top image seems to show two scale bars.
2. Western blotting showing images are not consistent in Fig.2, Fig.3 and Fig. S1.
3. Pay more attention of the tense in the figure legend, it is inconsistent, some are generally in the present tense, and some are in the past tense.
4. Correct spelling and capitalization (e.g. CITED2/Cited2, border/boarder,)
5. Verify that "E12.5 v E12.7" in Fig.1 should be "E12.5 v E17.5". E12.5 versus E17.4?
6. Spelling: "develomental" should be developmental; "indlunce" should be influence; "Alyernatively" should be Alternatively; "outgowth" should be outgrowth; "boarder" should be border; "stat site" should be start site; "anlayzed" should be analyzed; "Dextan-594" should be Dextran-594, "DGRs" should be DRGs, please double check and correct in the manuscript.
7. Formatting: "p0.07" should be p = 0.07; consistent capitalization of H3K9ac/H3K27ac; use "histone H3 acetylation" rather than "histone 3.", please check throughout the manuscript and correct the mislabeling.
8. For the analysis of the ATAC footprint, it is recommended to add TOBIAS/PIQ preference correction and motif enrichment validation.
9. In the quantification of images, it is recommended that authors clarify ROI, threshold, and double-blind methods; In the behavioral results of mice, it is recommended to clarify whether to randomize grouping, blinding, power analysis, and elimination criteria

Referee #2 (Remarks for Author):

This study reveals that CITED2 reverses the neuronal maturation process through epigenetic reprogramming, restoring the axonal regenerative capacity of mature dorsal root ganglion (DRG) neurons. The article first identifies a developmental-regeneration correlation window. It was discovered that the embryonic stage E12.5 (unpolarized stage) shares 190 upregulated

genes (e.g., related to transcriptional regulation and cell cycle) with regenerative injury (sciatic nerve axotomy, SNA), while the mature stage E17.5 (polarized stage) is enriched with downregulated synaptic-related genes. This suggests that the loss of regenerative capacity coincides with the polarization process.

CITED2 exhibits high expression at E12.5 and after SNA, but not after non-regenerative spinal cord dorsal column axotomy (DCA). Overexpression of CITED2 recruits EP300/CBP, increases H3K9ac/H3K27ac, opens chromatin (e.g., increased accessibility in 2615 enhancer regions), and activates developmental transcription factors such as bHLH and HOX (e.g., Meis2, HoxD13). This reprograms mature neurons to an E12.5-like state. Furthermore, the clinically approved HDAC inhibitor Panobinostat was found to promote axonal regeneration (its pro-growth effect disappears after Cited2 knockdown) and functional recovery (improved gridwalk test performance, increased 5-HT innervation) through a CITED2-dependent mechanism. The topic is very interesting and most experiments are properly done. However, I have some doubts on the presentation of the manuscript and on the conclusions reached by the authors based on these experiments. Additional data should be provided to sustain the discussion. Overall, the research showed plenty of phenotypes and data analysis related to neuronal maturation process and epigenetic reprogramming, however, the quality of the data in the article is not high and need improvements. And there are multiple format errors, especially in the data showing.

major comments:

Major comments:

1. What is the logic from Fig.1 to Fig.2? that means what is logic for narrowing down to Cited2? It seems that there is no correlation between data analysis and Cited2?
2. The image quality of Fig. 2 needs improvement. Panels are low-resolution and some are missing clear panel labels. Replace with higher-resolution images, ensure all panels are labeled consistently, and match figure numbers in text and legends.
3. In Fig.9, how to distinguish the newly regenerated axons from degenerating fibers. Better to use tracer injections or genetic lineage labeling to test. Or quantify axons by counts/length rather than fluorescence intensity ratios.
4. The mechanism by which Panobinostat elevates CITED2 expression remains unelucidated. While the paper mentions that Panobinostat increases histone acetylation, it does not validate whether this directly affects the CITED2 promoter.
5. CITED2 regulates cell cycle genes (e.g., CDK1) and is aberrantly expressed in tumors. The paper does not assess the potential negative effects of Panobinostat on other cells, such as whether it could cause abnormal proliferation or tumorigenesis?
6. Behavioral assessment strategy (Fig 9E), BBB/BMS scores are standard for SCI locomotor assessment, why the authors only did Gridwalk test? Better to test with BBB/BMS scores.
7. Here the author showing data that CITED2-overexpression upregulated a batch of genes, more accessible chromatin, like confirmed Cbx5 and HoxD3, however no direct evidence to show how these genes show more accessible chromatin? Could CITED2 bind directly and promote accessible chromatin?

Referee #3 (Comments on Novelty/Model System for Author):

In the study, the authors utilized both in vitro and in vivo models to comprehensively investigate the role of CITED2 in neuronal regeneration.

Referee #3 (Remarks for Author):

This study demonstrates that CITED2 is a critical transcriptional co-factor involved in maintaining axon regeneration potential in both immature DRG neurons and SNA-induced regenerating mature DRG neurons. Overexpression of CITED2 can reinitiate a growth-competent state in mature DRG neurons following DCA. Furthermore, the authors have elucidated the underlying mechanisms by which CITED2 promotes axonal regeneration, including the enhancement of chromatin accessibility and the induction of transcriptional changes reminiscent of early developmental stages. The findings suggest that the loss of regenerative capacity during neuronal maturation is an epigenetically regulated process that can be reversed through modulation of CITED2 expression. Importantly, the study identifies a clinically approved drug, Panobinostat, which enhances CITED2 expression and promotes neuronal regeneration both in vitro and in vivo, indicating its potential as a therapeutic strategy for enhancing neuronal regeneration. These findings underscore the significance of CITED2 in neuronal regeneration and provide novel avenues for therapeutic intervention in the field of neuroregenerative medicine. Congratulations on this valuable contribution from Prof. Di Giovanni's group.

I have a few minor comments:

- (1) The authors should explain why they chose only male mice for this study.
- (2) Given that CITED2 serves as a crucial molecular switch governing the transition of neurons from an immature to a mature state, it raises an important question regarding whether the long-term administration of Panobinostat to enhance CITED2 expression and promote neuronal regeneration may potentially affect neural structural stability.
- (3) A citation is missing on page 3, line 5.

REVIEW PROCESS FILE

Referee #1:

During development, axons grow over long distances to reach their postsynaptic targets. This growth capability is limited in time as reaching the postnatal stage, there is a shift from growth ability to connectivity (synapse formation and stabilization of the circuit). Most of adult axons do not grow anymore. However, in case of injury to the mature nervous system, inducing axon growth is key to establish and repair circuits. Therefore, it is critical to understand the molecular processes underlying such changes.

In the current work, Müller and colleagues compared the molecular state (RNA expression, chromatin state, promotor/enhancer accessibility) at key stages of neuronal development with neuronal state sustaining regeneration and when regeneration is abolished. To this end they used the very elegant model of dorsal root ganglia neurons. These neurons are very unique as they don't have any dendrites but present 2 axons: one projecting to the spinal cord, forming the dorsal column and the other one to the periphery and forming for example the sciatic nerve. While the peripheral branch is able to regenerate after an injury, the central branch has no regrowth ability.

Using this model, Authors show that when regeneration is induced (upon sciatic nerve injury), neuronal programs revert to a developmental state when axons are growing. This molecular plasticity is not triggered with dorsal column lesion. Interestingly one key factor underlying this mechanism is Cited-2 that is a transcription co-factor. Its manipulation in vitro and in vivo induces modulated regeneration. Indeed, its overexpression could induce dorsal column regeneration.

Interestingly, Cited-2 has been found as a target for Panobinostat as drug that could be used in cases of nervous system injury. Its administration in vivo is able to improve axon growth through regeneration and/or sprouting. These results lead to better functional outcome.

This work is very thorough and experiments are well executed.

We would like to thank the complimentary feedback of this reviewer and the questions

There are some minor comments/suggestions:

P4: "Importantly, this process is not irreversible": the double negation does not read well. Would it be possible to rephrase?

Great suggestion, this has been amended, thank you.

Could Authors explain the choice of dorsal column hemi section for the Panobinostat experiments instead of dorsal column crush as for the rest of the work?

Thanks for the question. Both models are widely accepted and appropriate for SCI research. Here is the explanation for our choice in this instance. Dorsal column over-hemisection provides a highly reproducible and complete interruption of ascending sensory axons with minimal spared fibres. By contrast, crush injuries often leave a variable number of spared axons, which can complicate interpretation of behavioural recovery especially when testing a drug (Panobinostat) expected to enhance sprouting and function. Further, functional recovery (e.g. sensory-motor coordination, proprioceptive tasks as analysed in the gridwalk) is easier to interpret in the dorsal over-hemisection model, since baseline deficits are more robust and consistent across animals. This makes it easier to attribute improvements more confidently to the drug intervention, rather than to variability in lesion severity.

Referee #2:

This study reveals that CITED2 reverses the neuronal maturation process through epigenetic reprogramming, restoring the axonal regenerative capacity of mature dorsal root ganglion (DRG) neurons. The article first identifies a developmental-regeneration correlation window. It was discovered that the embryonic stage E12.5 (unpolarized stage) shares 190 upregulated genes (e.g., related to transcriptional regulation and cell cycle) with regenerative injury (sciatic nerve axotomy, SNA), while the mature stage E17.5 (polarized stage) is enriched with downregulated synaptic-related genes. This suggests that the loss of regenerative capacity coincides with the polarization process.

CITED2 exhibits high expression at E12.5 and after SNA, but not after non-regenerative spinal cord dorsal column axotomy (DCA). Overexpression of CITED2 recruits EP300/CBP, increases H3K9ac/H3K27ac, opens chromatin (e.g., increased accessibility in 2615 enhancer regions), and activates developmental transcription factors such as bHLH and HOX (e.g., Meis2, HoxD13). This reprograms mature neurons to an E12.5-like state. Furthermore, the clinically approved HDAC inhibitor Panobinostat was found to promote axonal regeneration (its pro-growth effect disappears after Cited2 knockdown) and functional recovery (improved gridwalk test performance, increased 5-HT innervation) through a CITED2-dependent mechanism. The topic is very interesting and most experiments are properly done. However, I have some doubts on the presentation of the manuscript and on the conclusions reached by the authors based on these experiments. Additional data should be provided to sustain the discussion. Overall, the research showed plenty of phenotypes and data analysis related to neuronal maturation process and epigenetic reprogramming, however, the quality of the data in the article is not high and need improvements. And there are multiple format errors, especially in the data showing.

We would like to thank the very useful detailed review, the questions and suggestions.

Major comments:

1. What is the logic from Fig.1 to Fig.2? that means what is logic for narrowing down to Cited2? It seems that there is no correlation between data analysis and Cited2?

Thanks for the question. The goal of the study that was to identify critical epigenetic regulators of the transition from neuronal immature non-polarized to a mature polarized developmental stage that could underpin the loss of regenerative growth. CITED2 was the best candidate gene that met the criteria for been a developmentally regulated epigenetic factor as it had concomitant significant shift in gene expression, of chromatin accessibility and H3 acetylation, and had reliance on cohesin-dependent 3D chromatin architecture (Figure 2A). In addition, mRNA expression of Cited2 was observed to be highest at both an immature non-polarized E12.5 stage and following a regenerative competent SNA (Figure 2B). The text has been edited for clarity as well.

2. The image quality of Fig. 2 needs improvement. Panels are low-resolution and some are missing clear panel labels. Replace with higher-resolution images, ensure all panels are labeled consistently, and match figure numbers in text and legends.

This has been corrected, apologies and thank you.

3. In Fig.9, how to distinguish the newly regenerated axons from degenerating fibers. Better to use tracer injections or genetic lineage labeling to test. Or quantify axons by counts/length rather than fluorescence intensity ratios.

We did use tracer injection with Dextran. Additionally, we have added axon quantification by length to the previous fluorescence intensity as suggested (see Figure 9D).

4. The mechanism by which Panobinostat elevates CITED2 expression remains unelucidated. While the paper mentions that Panobinostat increases histone acetylation, it does not validate whether this directly affects the CITED2 promoter.

Thank you for raising this very important mechanistic question. We have now performed CUT&Tag for histone acetylation (H3k27ac) after Panobinostat and found that Panobinostat increased histone acetylation at the Cited2 promoter (see Figure EV5).

5. CITED2 regulates cell cycle genes (e.g., CDK1) and is aberrantly expressed in tumors. The paper does not assess the potential negative effects of Panobinostat on other cells, such as whether it could cause abnormal proliferation or tumorigenesis?

This is a very valid point, thank you for the question. We have not observed any tumour formation in our cords and mice did not show any adverse effects in the timeline investigated. Further, and perhaps more importantly Panobinostat has received FDA approval as adjuvant treatment against refractory multiple myeloma in patients, where it has proven to be safe and to limit tumorigenesis rather than inducing it.

6. Behavioral assessment strategy (Fig 9E), BBB/BMS scores are standard for SCI locomotor assessment, why the authors only did Gridwalk test? Better to test with BBB/BMS scores.

Thank you for the very pertinent comment. While BBB/BMS open-field scores are indeed widely used for assessing gross locomotor recovery after SCI, they are less sensitive to subtle deficits in skilled locomotion and fine motor control. Because our study was focused on sprouting and integration of sensorimotor pathways, these are more precisely and reliably measured by a test such as the Gridwalk, which is well-established for detecting subtle deficits in forelimb/hindlimb placement accuracy and recovery of skilled locomotion. Importantly, BBB/BMS scores are most informative for severe contusion or transection models where gross locomotor recovery is expected; in our dorsal column injury paradigm, the Gridwalk provides quantitative and reproducible measures (footslip errors) that more directly reflect the functional pathways under investigation.

7. Here the author showing data that CITED2-overexpression upregulated a batch of genes, more accessible chromatin, like confirmed Cbx5 and HoxD3, however no direct evidence to show how these genes show more accessible chromatin? Could CITED2 bind directly and promote accessible chromatin?

This is correct: we have not provided specific evidence, and we agree with this reviewer that CITED2 might bind directly and promote accessible chromatin. Therefore, we have added a sentence in the discussion explaining the possible mechanisms responsible for CITED2-dependent increase in accessibility including increase in acetylation of gene regulatory regions due to activation of CBP/p300 or direct CITED2 occupancy of gene regulatory regions.

Minor comments:

There are many errors in the manuscript, including spelling, tense, figure legends, and data analysis methods, and it must be double checked very seriously throughout the manuscript to ensure that all errors are corrected.

1. The image quality of Fig. 2 needs improvement. The figure legend misses the label "Fig 2". In Fig. 8C, the text "GFP" in the top image appears obscured. In Fig. 8E, the top image seems to show two scale bars.

2. Western blotting showing images are not consistent in Fig.2, Fig.3 and Fig. S1.

These issues have been amended, thank you, with apologies.

3. Pay more attention of the tense in the figure legend, it is inconsistent, some are generally in the present tense, and some are in the past tense.

This has been amended and made consistent, thank you

4. Correct spelling and capitalization (e.g. CITED2/Cited2, border/boarder,)

5. Verify that "E12.5 v E12.7" in Fig.1 should be "E12.5 v E17.5". E12.5 versus E17.4?

6. Spelling: "develontal" should be developmental; "indlunce" should be influence;

"Alyernatively" should be Alternatively; "outgowth" should be outgrowth; "boarder" should be

border; "stat site" should be start site; "anlayzed" should be analyzed; "Dextan-594" should be Dextran-594, "DGRs" should be DRGs, please double check and correct in the manuscript.

We would like to apologise, the word language spelling tool was mistakenly not active, and this has contributed to us missing these misspellings, thank you so much for picking these up. Cited2 when referring to gene and CITED2 to protein has now been used.

7. Formatting: "p0.07" should be $p = 0.07$; consistent capitalization of H3K9ac/H3K27ac; use "histone H3 acetylation" rather than "histone 3.", please check throughout the manuscript and correct the mislabeling.

Thank you, corrected with apologies.

8. For the analysis of the ATAC footprint, it is recommended to add TOBIAS/PIQ preference correction and motif enrichment validation.

Thank you, this has been done as suggested and figures amended. Worth mentioning that the message of the manuscript has remained unchanged.

9. In the quantification of images, it is recommended that authors clarify ROI, threshold, and double-blind methods; In the behavioral results of mice, it is recommended to clarify whether to randomize grouping, blinding, power analysis, and elimination criteria

These points have been clarified in the methods, elimination criteria in the checklist provided by the journal, thank you.

Referee #3:

This study demonstrates that CITED2 is a critical transcriptional co-factor involved in maintaining axon regeneration potential in both immature DRG neurons and SNA-induced regenerating mature DRG neurons. Overexpression of CITED2 can reinitiate a growth-competent state in mature DRG neurons following DCA. Furthermore, the authors have elucidated the underlying mechanisms by which CITED2 promotes axonal regeneration, including the enhancement of chromatin accessibility and the induction of transcriptional changes reminiscent of early developmental stages. The findings suggest that the loss of regenerative capacity during neuronal maturation is an epigenetically regulated process that can be reversed through modulation of CITED2 expression. Importantly, the study identifies a clinically approved drug, Panobinostat, which enhances CITED2 expression and promotes neuronal regeneration both in vitro and in vivo, indicating its potential as a therapeutic strategy for enhancing neuronal regeneration. These findings underscore the significance of CITED2 in neuronal regeneration and provide novel avenues for therapeutic intervention in the field of neuroregenerative medicine. Congratulations on this valuable contribution from Prof. Di Giovanni's group.

We would like to thank the complimentary feedback of this reviewer and the questions.

I have a few minor comments:

(1) The authors should explain why they chose only male mice for this study.

Thank you for the question. We focused on male mice for this SCI to minimize animal use while maintaining consistency and reproducibility in accordance with the 3Rs principles (Replacement, Reduction, and Refinement). Given the research questions did not specifically imply sex differences, male mice allowed us to avoid potential confounding effects of hormonal cycling on behavioural recovery metrics following SCI, which would have required a larger sample size. Male mice are also known to have poorer recovery after SCI and therefore they represent a greater challenge. That said, it would be useful to confirm these findings in both sexes in future studies.

(2) Given that CITED2 serves as a crucial molecular switch governing the transition of neurons from an immature to a mature state, it raises an important question regarding whether the long-term administration of Panobinostat to enhance CITED2 expression and promote neuronal regeneration may potentially affect neural structural stability.

In the timeline investigated we have not observed adverse effects, however, this is a very interesting question that would require proper dedicated studies with longer treatment regimens. We have added a sentence in the discussion to this effect.

(3) A citation is missing on page 3, line 5.

Thank you, this has now been added.

9th Jan 2026

Dear Prof. Di Giovanni,

Thank you for sending us your revised manuscript. We have now heard back from the reviewer who was asked to re-evaluate your study. As you will see, the reviewer is satisfied with the modifications made. Before we can formally accept your manuscript for publication, we would ask you to address the following editorial-level issues:

1. Please reduce the keyword number to five.
2. Remove "Authors' contribution" section from the manuscript file.
3. The funding information appears incomplete in the submission system. Please add "Wings for Life; Spinal Research; Brain Research Trust, the Dr. Miriam and Sheldon G Adelson Medical Research Foundation and the National Institute for Health Research (NIHR) Imperial Biomedical Research Centre" to the list of funders in our system. Also, please add project numbers where available.
4. The paper explained: EMBO Molecular Medicine articles are accompanied by a summary of the articles to emphasize the major findings in the paper and their medical implications for the non-specialist reader.

Please provide a summary of your article highlighting

Please refer to any of our published articles for an example.

5. Appendix:

- please add page numbers to the items listed in the Table of Content.
- Remove the entire section " Supplementary Datasets" from the Table of Content.
- Rename " Table EV1" to "Appendix Table S1" and update the callout accordingly.

6. Please provide a complete author checklist, which you can download from our author guidelines (<https://media.springernature.com/original/springer-cms/rest/v1/content/27825796/data/v1>). Please insert information in the checklist that is also reflected in the manuscript. The completed author checklist will also be part of the RPF.

7. Please download and fill our Reagents and Tools Table template (.docx), which you can find in our author guidelines: <https://link.springer.com/journal/44320/submission-guidelines#structuredmethods>

8. EV datasets: please add the legend to each corresponding excel file, in a separate tab/worksheet. Remove Dataset EV8 and its callout.
9. BIORENDER usage: please remove from Acknowledgments and add a dedicated section to the Methods using this format:

Graphics:

(some of the... OR Figure #... OR synopsis) Graphics were created with BioRender.com.

10. Please remove the visual abstract/synopsis image from the manuscript text and upload it as a separate PNG file.

11. Please provide a 'Synopsis' (as a separate file) to further enhance discoverability. Synopses are displayed on the journal webpage and are freely accessible to all readers. They include a short stand first (maximum of 300 characters, including space) as well as 2-5 one-sentences bullet points that summarizes the paper. Please write the bullet points to summarize the key NEW findings. They should be designed to be complementary to the abstract - i.e. not repeat the same text. We encourage inclusion of key acronyms and quantitative information (maximum of 30 words / bullet point).

12. Data availability: the specific URLs for GSE214722, GSE224110 datasets should be provided in the data availability statement.

13. Please add missing callouts for Appendix Figure S5 and S7; please correct the callouts for Figure S11 to "Appendix Figure S11".

14. Source Data:

- Source data for Fig. 1C, D, E; Fig. 5B, I, J, K; Fig. 6B, E, F; and Fig. 7 are not required. Please untick these panels in the source data checklist.

- Dataset EV8 contains numerical source data for multiple figure panels. Please combine the numerical data with the corresponding imaging data for each figure and upload them together as a single zipped source data (SD) file per figure.

15. During our standard image screening, our data integrity officer identified the following issues:

- Figure 4E and Appendix Figure S3C (AAV-GFP): Possible image reuse was noted. Any reuse must be explicitly disclosed and clearly indicated in the figure legend.

- Figure EV2: A potential editing line was observed in the lower left panel. Please provide the original source data for this panel.

16. Data citation: In the Reference list, data citations must be labeled with "[DATASET]". A data reference must provide the database name, accession number/identifiers and a resolvable link to the landing page from which the data can be accessed at the end of the reference. Further instructions are available at < <https://link.springer.com/journal/44320/submission-guidelines#cms-Reference-guidelines> >.

17. Please address the following issues in figure legends:

- Please note that the legend for figure EV3 is missing in the manuscript. This needs to be rectified.

- Please define the annotated p values ****/**/*/* as well as provide the exact p-values for the same in the legend of figure 2D, G, J; 3B, D, F, H, J; 4C, D, F; 8B, D, F; 9C, D, F, H; EV1 A, C, E; EV5 appropriate.

- Please note that the exact p values are not provided in the legends of figures 5H, EV4 A,

- Please indicate the statistical test used for data analysis in the legends of figures 1E, 5I, 6C, D, E, F; 7E, F; EV4 B

- Please note that information related to n is missing in the legends of figures 7E, F

- Please note that the error bars are not defined in the legends of figures 2D, F, G, J; 3B, D, F, H, J; 4C, D, F; 8B, D, F; 9C, D, E, F, H; EV1 A, C, E; EV5

18. Please correct the headings and order of the sections to: Abstract/ Keywords / The Paper Explained / Introduction / Results / Discussion / Methods / Data Availability / Acknowledgements / Disclosure and Competing Interests Statement / References / Figure Legends / Expanded View Figure Legends

I look forward to receiving a revised version of your manuscript as soon as possible.

Kind regards,
Jingyi

Jingyi Hou
Senior Editor
EMBO Molecular Medicine

***** Reviewer's comments *****

Referee #2 (Comments on Novelty/Model System for Author):

-

Referee #2 (Remarks for Author):

The results in this study underscore CITED2 in neuronal regeneration and provide novel avenues for therapeutic intervention in the field of neuroregenerative medicine. It's a very interesting topic. Overall, based on the comments of all the reviewers, the quality of the data in the article is have been improved. The format errors, especially in the data showing, have corrected. The authors have addressed the questions I concerned. Suggest acceptance and format the version.

The authors addressed the remaining editorial issues.

29th Jan 2026

Dear Prof. Di Giovanni,

We are pleased to inform you that your manuscript is accepted for publication and is now being sent to our publisher to be included in the next available issue of EMBO Molecular Medicine.

You may qualify for financial assistance for your publication charges - either via a Springer Nature fully open access agreement or an EMBO initiative. Check your eligibility: <https://link.springer.com/journal/44321/how-to-publish-with-us>

Yours sincerely,
Jingyi

Jingyi Hou
Senior Editor
EMBO Molecular Medicine

>>> Please note that it is EMBO Molecular Medicine policy for the transcript of the editorial process (containing referee reports and your response letter) to be published as an online supplement to each paper. If you do NOT want this, you will need to inform the Editorial Office via email immediately. More information is available here: <https://link.springer.com/partners/embo-press/editorial-policies#Peer%20review>